# AI-enabled virtual spatial proteomics from histopathology for interpretable biomarker discovery in lung cancer

Spatial proteomics enables high-resolution mapping of protein expression and can transform our understanding of biology and disease. However, major challenges remain for clinical translation, including cost, complexity and scalability. Here we present H&E to protein expression (HEX), an AI model designed to computationally generate spatial proteomics profiles from standard histopathology slides. Trained and validated on 819,000 histopathology image tiles with matched protein expression from 382 tumor samples, HEX accurately predicts the expression of 40 biomarkers encompassing immune, structural and functional programs. HEX demonstrates substantial performance gains over alternative methods for protein expression prediction from H&E images. We develop a multimodal data integration approach that combines the original H&E image and AI-derived virtual spatial proteomics to enhance outcome prediction. Applied to six independent non-small-cell lung cancer cohorts totaling 2,298 patients, HEX-enabled multimodal integration improved prognostic accuracy by 22% and immunotherapy response prediction by 24–39% compared with conventional clinicopathological and molecular biomarkers. Biological interpretation revealed spatially organized tumor–immune niches predictive of therapeutic response, including the co-localization of T helper cells and cytotoxic T cells in responders, and immunosuppressive tumor-associated macrophage and neutrophil aggregates in non-responders. HEX provides a low-cost and scalable approach to study spatial biology and enables the discovery and clinical translation of interpretable biomarkers for precision medicine.

Spatial biology represents a new frontier of fundamental discovery and precision medicine[1-4]. Spatial molecular profiling is being used to reveal novel insights about disease biology, as well as mechanisms of response and resistance to therapies[5-9]. Technologies for spatial proteomics have evolved over the past decade, generating highly multiplexed images of specimens with increasing depth and resolution[10]. For instance, co-detection by indexing (CODEX) can identify and quantify more than 50 protein biomarkers simultaneously at single-cell resolution across entire tissue sections[11]. However, despite these technological advances, significant challenges remain for clinical translation, including cost, complexity and scalability[2]. In addition, workflows for spatial proteomics require specialized high-end instruments and trained personnel, further complicating clinical adoption.

Histopathology with hematoxylin and eosin (H&E) staining is the gold standard for clinical diagnosis of cancer and other diseases. Recent advances in deep learning have made it possible to estimate specific

✉e-mail: rli2@stanford.edu

molecular features, such as genetic mutations and gene expression, from H&E images[12–16]. An important limitation of previous studies is that model training is performed at the bulk tissue level[17]. This creates both a technical challenge, requiring weakly supervised learning from limited samples, and a fundamental challenge, due to the presence of intratumor heterogeneity. Spatial proteomics offers a promising solution to this problem by correlating tissue morphology with spatially resolved protein expression.

The feasibility of predicting spatial protein expression from histology images has been demonstrated. Previous efforts relied on unpaired or paired H&E and immunohistochemistry (IHC)[18,19], paired IHC and multiplexed immunofluorescence[20], or autofluorescence and multiplexed immunofluorescence images[21]. Given the inherent limitations of traditional multiplexed imaging assays, these studies are focused on only a few protein biomarkers. Importantly, the translational value has not been established without extensive clinical validation for predicting treatment outcomes on large cohorts of patients for whom only H&E images are available.

Here we present H&E to protein expression (HEX), an AI model designed to computationally generate high-dimensional spatial proteomics profiles directly from standard histopathology slides. HEX leverages a pathology foundation model to predict the expression of 40 protein biomarkers spanning immune, structural and functional programs, demonstrating substantial performance gains over alternative computational methods. We evaluate the prognostic and predictive relevance of HEX in multi-institutional cohorts of 2,298 patients with non-small-cell lung cancer (NSCLC). HEX significantly improves the accuracy of prognosis prediction in early-stage lung cancer compared with clinical risk factors. The prognostic utility of HEX is further confirmed across 12 additional cancer types involving 5,019 patients. Finally, HEX identifies spatial patterns of tumor–immune niches that improve the prediction of response to immune checkpoint blockade. HEX provides a low-cost and scalable approach to study spatial biology and facilitates its clinical translation to advance precision medicine.

## Results

### Study overview

We performed CODEX and H&E staining of the same tissue section for tumor samples from ten patients with NSCLC. For CODEX, we designed a panel of 40 proteins spanning lineage, immune, structural and functional markers, enabling comprehensive spatial phenotyping of the tumor microenvironment (Fig. 1a). The whole-slide images (WSIs) from the two experiments were co-registered and cropped into smaller image tiles measuring ~50 μm. Through this process, we obtained approximately 755,000 image tiles with 40 protein biomarkers and matched H&E histopathology, which formed the training dataset. HEX was trained by leveraging state-of-the-art pathology foundation models[22–25] to predict the expression of 40 protein biomarkers simultaneously based on H&E images, enabling the generation of virtual spatial proteomics profiles from standard histopathology (Fig. 1a and Extended Data Fig. 1). We assessed the model accuracy for protein

expression on two independent datasets with CODEX and H&E-stained tissue sections for 372 tumor samples. To further evaluate HEX's generalizability beyond NSCLC, we externally validated HEX on a pan-cancer dataset containing 57-plex CODEX and matched H&E images across 206 tumor samples from 34 tissue types[11].

We further developed a multimodal data integration framework combining the original H&E image and AI-derived virtual spatial proteomics to enhance outcome prediction (Fig. 1b). The clinical relevance of HEX was evaluated across five independent NSCLC cohorts with 2,150 patients and 12 other cancer types with 5,019 patients for prognosis prediction, as well as in a separate cohort of 148 patients treated with immune checkpoint inhibitors (ICIs) for response prediction. A flow chart of sample inclusion and exclusion criteria across the six lung cancer cohorts is provided in Extended Data Fig. 2. Finally, we aimed to provide biological interpretation of our results—linking the predicted outcomes with virtual spatial proteomics profiles—to identify tumor–immune spatial niches associated with therapy response and resistance. The following sections detail the technical performance of HEX in generating virtual spatial proteomics, and demonstrate the clinical relevance and biological insights derived from this approach in NSCLC.

### HEX enables accurate prediction of protein expression from H&E images

We evaluated the performance of HEX for protein prediction using cross-validation and independent validation, as shown in Figs. 2 and 3. For a comprehensive evaluation, we computed four complementary metrics: Pearson's correlation coefficient ($r$), Spearman's $r$, the structural similarity index measure (SSIM)[26] and mean squared error (MSE). We benchmarked HEX against two generative adversarial network (GAN)-based methods previously used for histological image translation[18,20].

**Cross-validation performance.** We performed fivefold cross-validation on the Stanford-WSI dataset, randomly splitting patients into training (80%) and validation (20%) sets in each fold to prevent patient-level data leakage. Across 40 biomarkers, HEX achieved accurate prediction of protein expression from H&E images with an average Pearson's $r$ of 0.790, a Spearman's $r$ of 0.787, an SSIM of 0.949 and an MSE of 0.076 (Fig. 2a). HEX consistently outperformed both GAN methods, with substantial improvements of 46% in Pearson's $r$, 44% in Spearman's $r$, 15% in SSIM and an 80% reduction in MSE compared with the second-best model, conditional GAN (CGAN)[20] (Fig. 2b).

To qualitatively assess spatial patterns of the predicted biomarkers, we visualized the true CODEX along with virtual CODEX generated from the H&E images. The HEX model produces high-fidelity virtual CODEX images, accurately reflecting the spatial organization and expression patterns of structural, immune, lineage and functional markers (Fig. 2c and Extended Data Fig. 3). It is worth noting that HEX has the capability of generating virtual CODEX images at any user-specified spatial resolution. To demonstrate this, we trained

---

**Fig. 1 | Development, validation and clinical applications of HEX.**
**a**, Development and technical validation of HEX, a predictive AI model designed to generate high-dimensional spatial proteomics profiles from standard H&E-stained slides. HEX was trained on a dataset of over 755,000 image tiles with matched protein expression, derived from tumor samples co-stained with H&E and high-plex immunofluorescence using a 40-antibody panel in ten patients with NSCLC. HEX leverages a state-of-the-art pathology foundation model to predict the expression of 40 protein biomarkers simultaneously based on H&E images, enabling the generation of virtual spatial proteomics profiles from standard histopathology. Model accuracy for protein expression was validated on two independent datasets with co-stained tissue sections for 372 tumor samples. To evaluate generalizability, HEX was validated externally on a pan-cancer dataset comprising 206 tumor samples across 34 tissue types, imaged

with 57-plex CODEX and matched H&E sections prepared using distinct staining protocols and scanners. **b**, Clinical validation and biological interpretation. The clinical relevance of HEX was evaluated for prognosis prediction across five NSCLC cohorts comprising 2,150 patients, as well as 5,019 patients from TCGA with 12 additional cancer types. It was also evaluated for response prediction in a separate cohort of 148 patients treated with ICIs. We developed a multimodal data integration method that combines the original H&E image and AI-derived virtual spatial proteomics profiles to further enhance outcome prediction. The HEX-generated biomarker distributions reveal spatial patterns of tumor–immune niches, providing biological rationale and insights into the prediction of clinical outcomes. FFPE, formalin fixed and paraffin embedded. Schematic elements created with BioRender.com.

**a**  AI-enabled generation of spatial proteomics from histopathology

(1) Experimental procedure and data acquisition

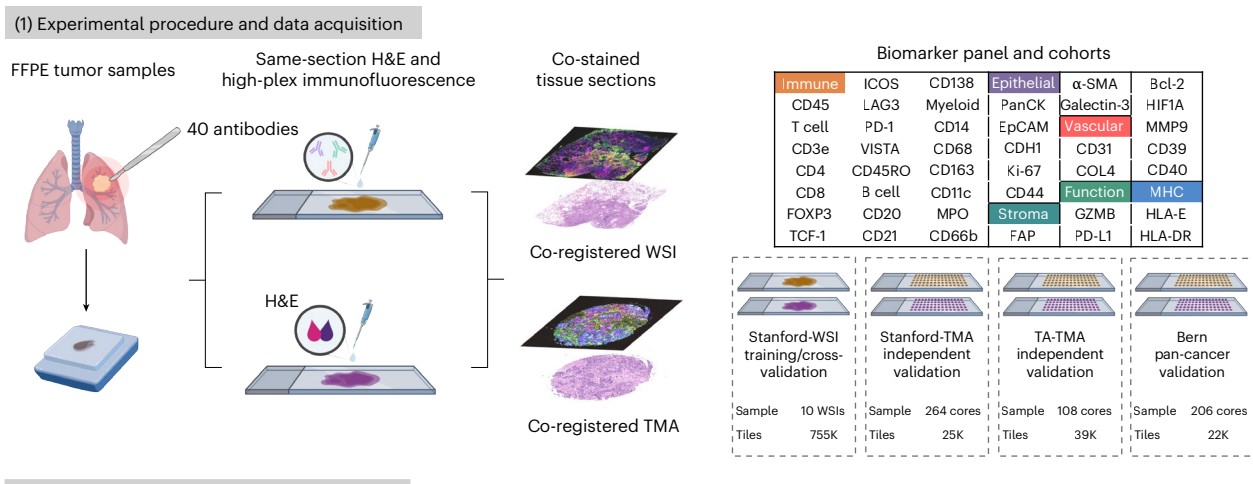

(2) Virtual CODEX from routine histopathology

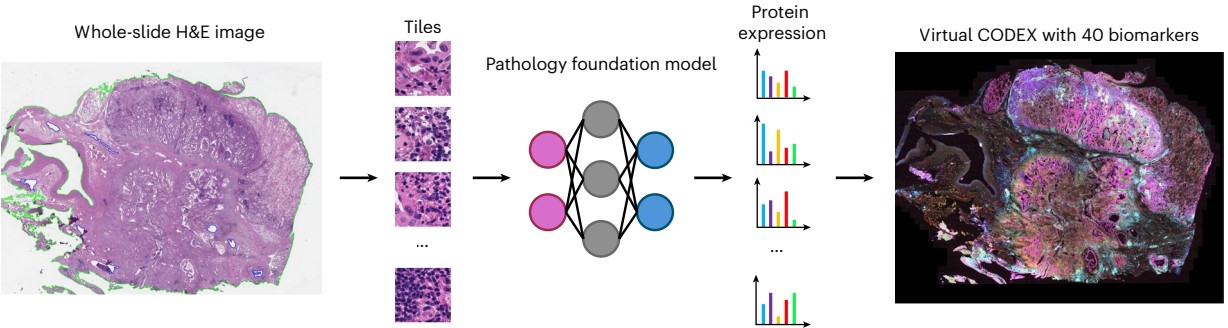

**b**  Multimodal data integration for the prediction of prognosis and immunotherapy response

(1) Model training and validation

(2) Patient cohorts

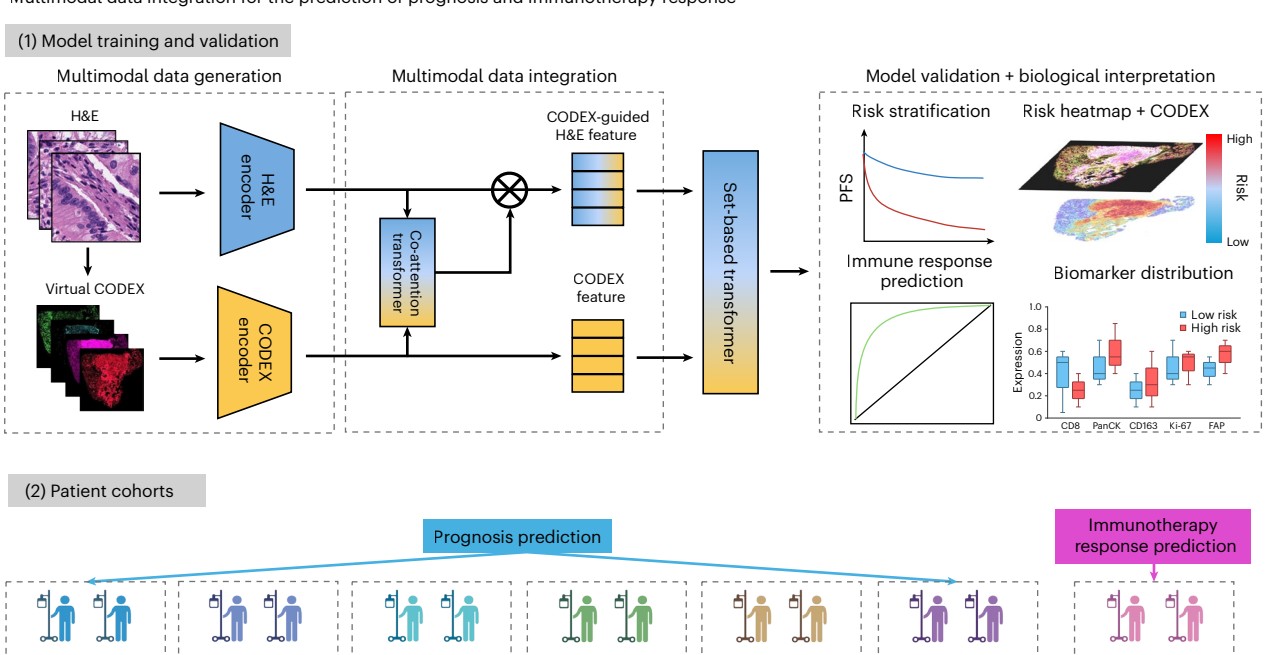

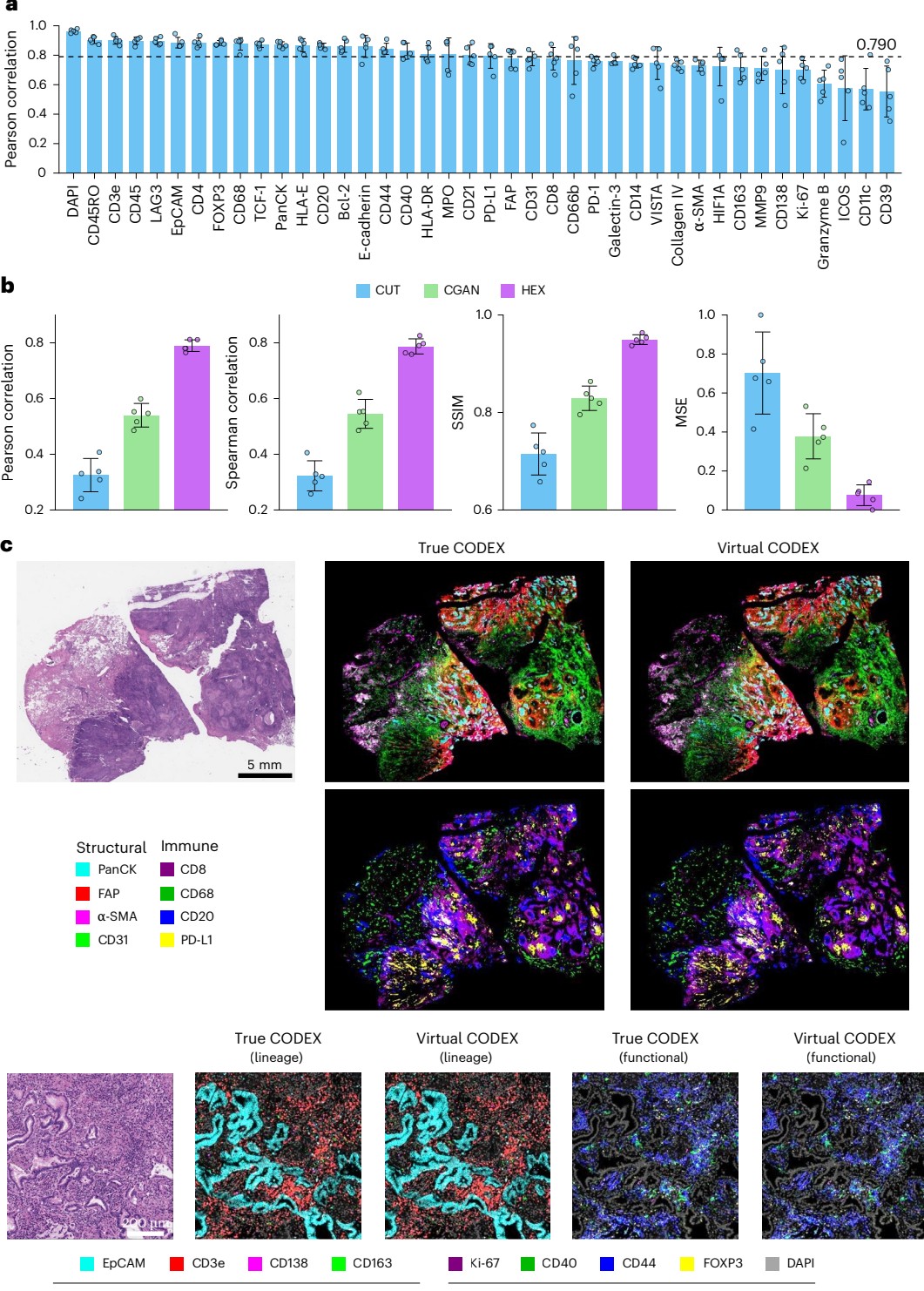

**Fig. 2 | Performance evaluation of HEX for protein biomarker prediction.**
**a**, Performance of HEX for predicting 40 protein biomarkers in cross-validation using ten WSIs. The results demonstrate high concordance between measured and predicted biomarker distributions. **b**, Comparison of HEX with two generative AI models using various performance metrics. **c**, Representative visualizations of HEX-predicted virtual spatial proteomics profiles. The virtual CODEX images demonstrate high resemblance to the true images across structural, immune, lineage and functional biomarkers at both standard and fine spatial resolutions, with 224- and 14-pixel patch sizes, respectively. Similar results were obtained in ten independent WSIs. In **a** and **b**, the bars represent means across fivefold cross-validation on the Stanford-WSI dataset ($n = 10$ WSIs), the data points represent individual folds and the error bars indicate standard deviation.

HEX with smaller window and step sizes (from 224 down to 14 pixels), enabling high-resolution virtual CODEX outputs and enhanced visualization of fine-grained spatial patterns (Fig. 2c, Extended Data Fig. 3 and Methods).

**Independent validation performance.** To test the generalizability of HEX, we trained the model on the full Stanford-WSI dataset and then evaluated its performance on two independent tissue microarray (TMA) cohorts—Stanford-TMA and tissue array TMA (TA-TMA)—comprising

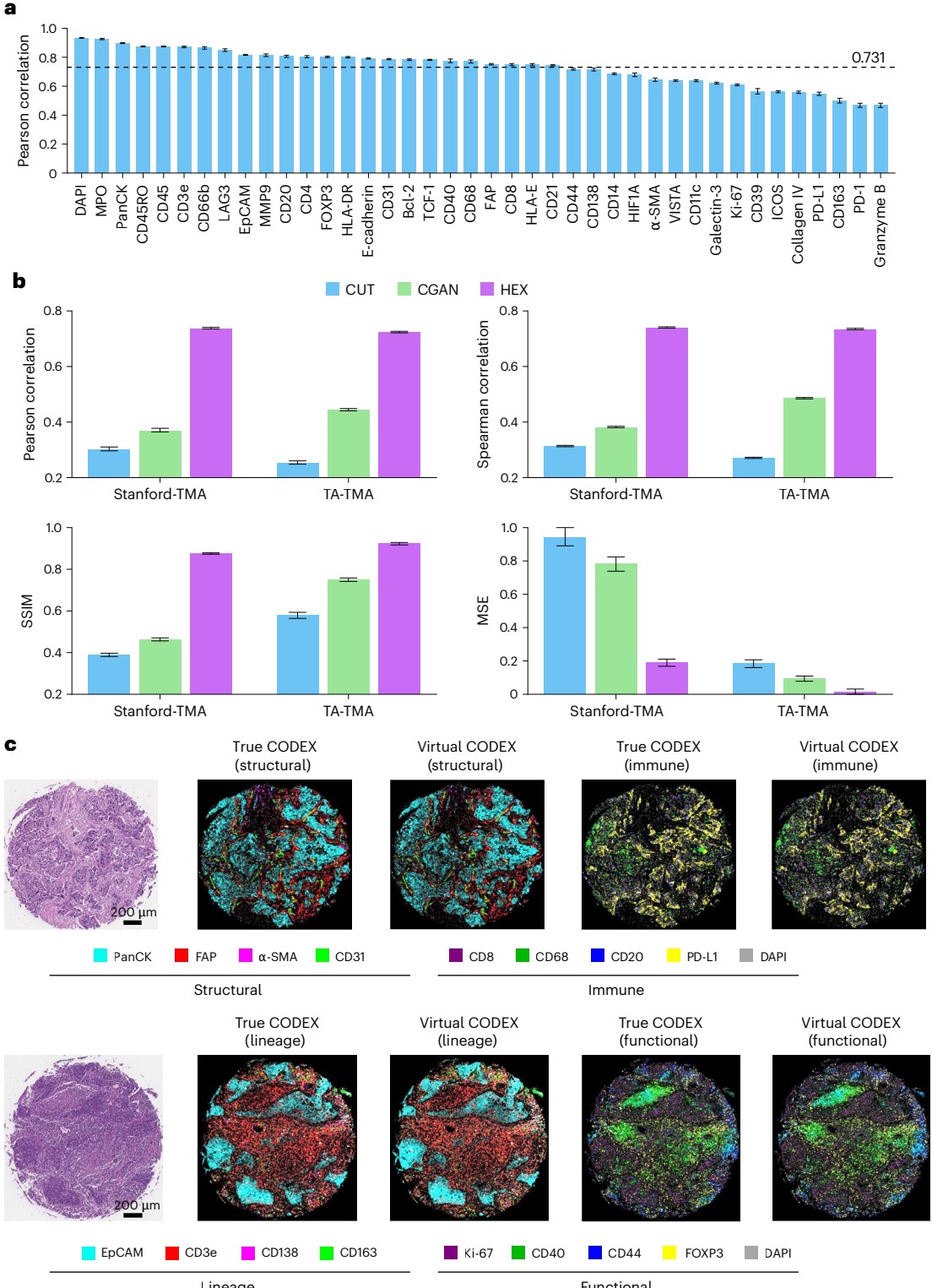

**Fig. 3 | Independent validation of HEX for protein biomarker prediction.**
**a**, Performance of HEX for predicting 40 protein biomarkers across independent
test sets comprising 372 TMA cores (Stanford-TMA ($n = 264$) and TA-TMA
($n = 108$)). **b**, Comparison of HEX with two generative AI models using various
performance metrics. **c**, Representative visualizations of HEX-predicted
virtual spatial proteomics profiles. The virtual CODEX images demonstrate
high resemblance to the true images across structural, immune, lineage and
functional biomarkers. Visualizations were generated using the high-resolution
version of HEX with a 14-pixel patch size to enhance the spatial details. Similar
results were obtained in 372 cores. In **a** and **b**, the bars represent point estimates
and the error bars indicate 95% bootstrap CIs ($n = 1,000$ resamples).

372 tumor samples in total. HEX showed consistently high Pearson's
*r* values across all 40 biomarkers, as shown in Fig. 3a. On the first
Stanford-TMA dataset, HEX achieved an average Pearson's *r* of 0.738,
Spearman's *r* of 0.741, SSIM of 0.875 and MSE of 0.189 across all

biomarkers. Compared with the second-best model (CGAN), HEX
nearly doubled the key performance metrics, with a Pearson's *r* of 0.738
versus 0.370, a Spearman's *r* of 0.741 versus 0.382, an SSIM of 0.875
versus 0.463 and a 76% reduction in MSE (0.189 versus 0.782) (Fig. 3b).

HEX demonstrated similarly strong prediction performance in the second TMA dataset. Together, these results highlight HEX's generalizability and robustness across independent datasets, with HEX substantially outperforming existing methods across various evaluation metrics. We further stratified performance by subcellular localization (nuclear, cytoplasmic and membrane) and observed comparable accuracy across groups (Supplementary Results and Extended Data Fig. 4).

We also assessed the quality of virtual CODEX images generated from H&E images on the independent TMA datasets. As both datasets comprise relatively small cores, we employed the high-resolution variant of HEX with smaller window and step sizes, to enhance the visualization of biomarker distributions. Representative images including structural, immune, lineage and functional biomarkers are shown in Fig. 3c.

To assess the generalizability of HEX to new tissue types and its robustness to variations in histological protocols, we conducted external validation using a publicly available pan-cancer dataset[11] containing 57-plex CODEX and H&E images of the same section. This dataset comprises 206 TMA cores spanning 34 distinct tissue types, including malignant tumors (for example, breast, colorectal, liver, pancreas, kidney and bladder), benign neoplasms and matched normal tissues. The samples were obtained from the University of Bern in Switzerland, prepared using a different H&E staining protocol and digitized using a Keyence BZ-X710 scanner. Despite these differences, HEX achieved strong predictive performance across 24 overlapping biomarkers without fine-tuning (mean Pearson's $r$: 0.658)—only slightly lower than on the Stanford dataset (mean Pearson's $r$: 0.718) (Extended Data Fig. 5). Furthermore, HEX substantially outperformed the second-best model (CGAN) across all evaluation metrics: Pearson's $r$ (0.658 versus 0.210), Spearman's $r$ (0.563 versus 0.140), SSIM (0.638 versus 0.521) and MSE (0.132 versus 0.835). These results demonstrate HEX's robustness and generalizability across diverse tissue types, staining protocols and imaging platforms.

To evaluate HEX's adaptability to new tissue types and expanded biomarker panels, we conducted both retraining and fine-tuning experiments on 140 colorectal cancer (CRC) cores from the Bern dataset, using all 57 protein markers—33 of which were not present in the original NSCLC panel. The same data split was used for both experiments: 84 cores from 21 patients for training and 56 cores from 14 patients for testing. When retrained from scratch, HEX achieved good predictive performance (mean Pearson's $r$: 0.566; Extended Data Fig. 6), demonstrating its ability to generalize to new protein markers in a different tissue. Fine-tuning the NSCLC-trained HEX model—by re-initializing the output head and adapting to CRC data—further improved the performance (mean Pearson's $r$: 0.659; Extended Data Fig. 7). Notably, improvements were observed for both shared and CRC-specific biomarkers (Extended Data Fig. 8). These results confirm that HEX can be efficiently adapted to new tissues and marker panels with minimal architectural changes and limited training data, enabling scalable deployment of virtual spatial proteomics across diverse histological contexts.

We next performed orthogonal validation of HEX-derived spatial proteomics using matched IHC and H&E slides[27]. Evaluated on three lung tumors, HEX predicted IHC measurements with good spatial concordance—Pearson's $r$ = 0.479 (CD31) and 0.606 (Ki-67)—and substantially outperformed the CGAN baseline (Pearson's $r$: 0.542 versus 0.086; Extended Data Fig. 9).

We finally evaluated the impact of using different foundation model backbones and training strategies. HEX models initialized from the MUSK[22] backbone achieved the highest overall accuracy, whereas models based on the CONCH[24] backbone showed faster inference at the expense of lower accuracy (Supplementary Figs. 1 and 2). To dissect the role of training strategies, we conducted ablation experiments by removing either feature distribution smoothing (FDS) or adaptive loss function (ALF) from the original HEX model. In both cases, model performance decreased notably across all metrics, indicating that these components are necessary for achieving robust and generalizable predictions (Supplementary Figs. 3 and 4).

## HEX improves prognosis prediction in early-stage lung cancer

With the ability to generate spatial proteomics from standard H&E images, HEX enables biologically interpretable prediction of clinically relevant outcomes by adding a new layer of molecular insight. Although H&E provides detailed tissue morphology, virtual CODEX maps offer complementary information about spatially resolved protein expression. To integrate these distinct yet synergistic data types, we developed multimodal integration via co-attention (MICA), a deep learning framework that fuses H&E and virtual CODEX data at an early stage (Supplementary Fig. 5 and Methods). This approach explicitly models cross-modal interactions and spatial relationships, enhancing its ability to identify clinically relevant features predictive of patient outcomes.

We evaluated the prognostic utility of MICA-based risk predictions across five independent NSCLC cohorts. Using data from the National Lung Screening Trial (NLST)[28], we trained a model to predict recurrence-free survival (RFS) from H&E-stained WSIs and then tested in four other patient cohorts, including The Cancer Genome Atlas (TCGA), Prostate, Lung, Colorectal, and Ovarian (PLCO)[29], Stanford-TMA and TA-TMA cohorts. Kaplan–Meier analyses demonstrated significant and robust risk stratification across the training and validation cohorts (Fig. 4a). The hazard ratio (HR) between MICA-defined high- and low-risk patients was 2.43–3.33 (all $P \le 0.002$) for stage I NSCLC and 2.27–4.28 (all $P \le 0.012$) for stage II NSCLC in the validation cohorts. Similar results were observed in the TA-TMA cohort for overall survival (Supplementary Fig. 6).

Quantitative evaluation using the concordance index (C-index) further confirmed the MICA risk model's prognostic performance (Fig. 4b). For patients with early-stage disease, MICA achieved C-indices of 0.80 (NLST), 0.67 (TCGA), 0.68 (Stanford-TMA), 0.72 (PLCO) and 0.62 (TA-TMA). Across the four validation cohorts, MICA attained an overall adjusted C-index of 0.68 (weighted by the number of patients), significantly outperforming the H&E-only model (overall C-index: 0.56; all $P \le 0.016$) and virtual CODEX-only model (overall C-index: 0.59).

The MICA risk model also outperformed traditional clinicopathological variables, including tumor stage and grade for prognosis prediction in early-stage NSCLC (Fig. 4c). Multivariable Cox regression analyses further confirmed MICA-derived risk scores as the strongest independent predictor of clinical outcomes (RFS, disease-specific survival (DSS) and overall survival) across all patient cohorts (Fig. 4d). Importantly, integrating MICA-derived scores with standard risk factors such as age, sex, grade and stage significantly enhanced the prognostic accuracy compared with clinical variables alone (overall C-index: 0.71 versus 0.58; all $P \le 0.002$) in early-stage NSCLC. The substantial increase of 22% in prognostic accuracy underscores the value of fusing H&E images and virtual spatial proteomics, providing complementary information beyond traditional clinicopathological risk factors.

Additional analyses confirmed robustness across disease stages and clinical subgroups (Supplementary Figs. 7–12), per-cohort cross-validation (Supplementary Fig. 13) and against a late-fusion baseline (PORPOISE[30]) in both cross-validation and independent validation (Supplementary Results and Supplementary Fig. 14). Moreover, across 12 TCGA cancer types, MICA outperformed both unimodal baselines under fivefold cross-validation (Supplementary Results and Supplementary Fig. 15).

To compare the prognostic value of virtual versus experimentally acquired spatial proteomics, we applied the NLST-trained models to two independent cohorts (Stanford-TMA and TA-TMA) with matched H&E and true CODEX data. We compared three models: (1) an H&E-only baseline; (2) MICA using HEX-generated virtual CODEX (MICA-Virtual); and (3) MICA using true CODEX (MICA-True). As shown in Supplementary Fig. 16, both MICA-Virtual and MICA-True models significantly outperformed the H&E-only model in prognostic

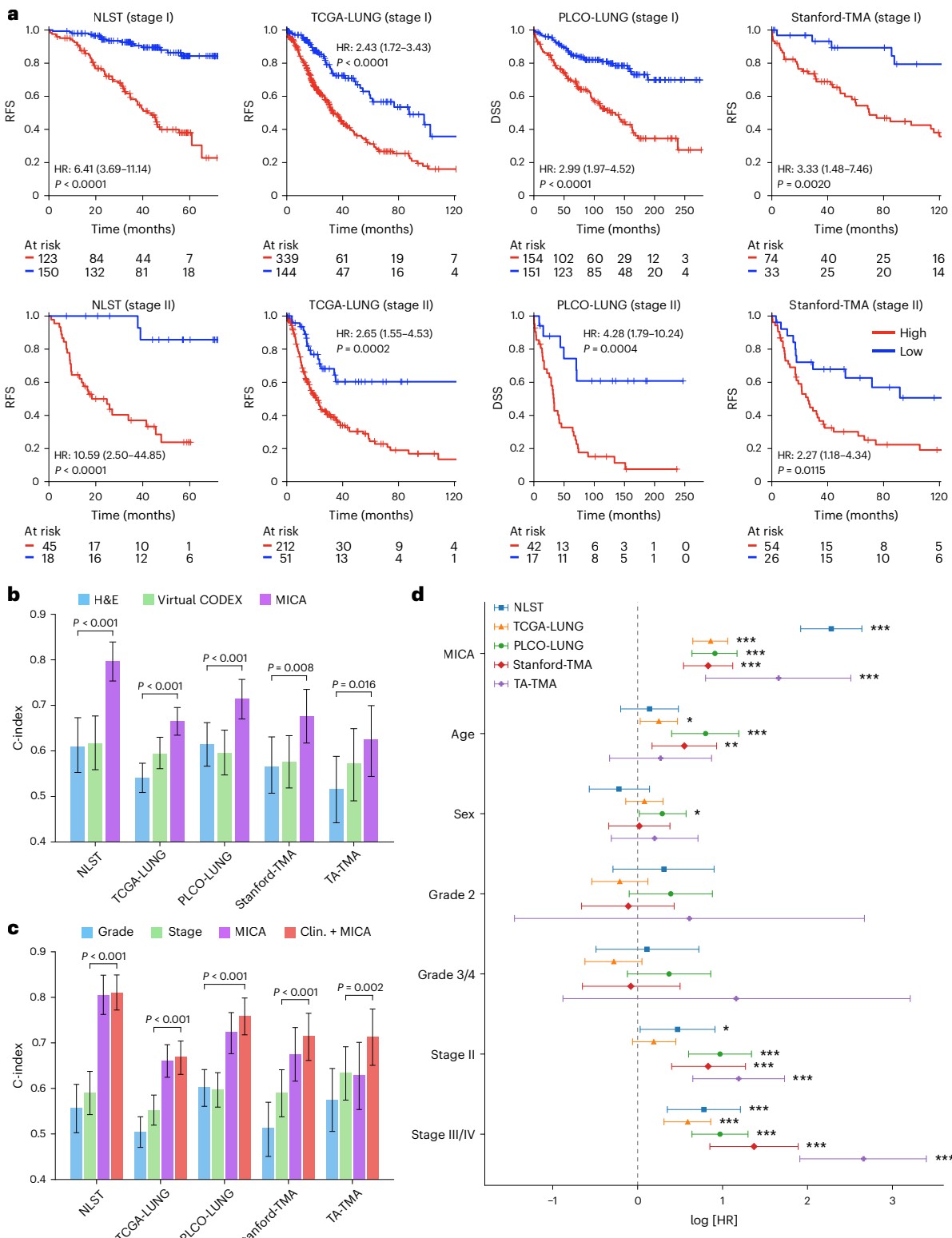

**Fig. 4 | MICA improves prognosis prediction in patients with early-stage NSCLC.** **a**, Kaplan–Meier analysis of MICA-predicted risk groups in patients with stage I and II NSCLC. Patients were significantly stratified into high- and low-risk groups (red and blue, respectively) for RFS or DSS across four independent cohorts. All cutoffs were the same across validation cohorts and determined based on the NLST training cohort. The numbers of patients at risk for each group and time point are given below each graph. Statistical significance was determined by two-sided log-rank test. **b**, Performance comparison of prognostic models across five cohorts of early-stage NSCLC. **c**, Performance comparison of MICA risk prediction against clinical features (Clin.) in early-stage NSCLC. In **b** and **c**, the bars represent point estimates and error bars indicate 95% bootstrap CIs (n = 1,000 resamples). **d**, Multivariable Cox regression analysis incorporating clinical covariates and MICA-predicted risk scores across five cohorts (NLST (n = 336 patients), TCGA-LUNG (n = 746), PLCO-LUNG (n = 364), Stanford-TMA (n = 187) and TA-TMA (n = 94)). The points denote log[HR] point estimates and the horizontal bars represent 95% CIs. The MICA risk score remains a significant independent predictor of RFS, DSS or overall survival. Statistical significance was determined by two-sided Wald test and P values were unadjusted. ***P < 0.001; **P < 0.01; *P < 0.05.

accuracy. Importantly, there was no statistically significant difference in C-index between MICA-True and MICA-Virtual in either cohort. These results demonstrate that HEX-derived virtual CODEX can provide additional prognostic information beyond H&E that is equivalent to true CODEX data.

## Biological interpretation of HEX-based prognosis prediction

Our HEX approach not only enables accurate prognosis prediction but can also provide biological interpretation behind the predictions. Figure 5a shows the virtual spatial proteomics profiles generated from standard H&E images for selected key biomarkers that correspond to model-predicted high- and low-risk patients. This allows us to link tissue morphology with spatially resolved protein expression and pinpoint regions of interest that contribute most strongly to patient risk.

To assess the biological relevance of model predictions, we applied the integrated gradients method[31] to compute tile-level attribution scores. We then categorized image tiles with the top and bottom integrated gradient values across the NLST cohort—representing the most influential tiles contributing to the model's risk predictions—into high- and low-risk groups, respectively (Fig. 5b and Methods). Morphologically, we observed histologic patterns such as poor differentiation, lymphovascular invasion, necrosis and micropapillary patterns among high-risk tiles. In contrast, low-risk tiles included patches with lepidic growth, well-differentiated tumor, abundant tumor-infiltrating lymphocytes and tertiary lymphoid structures—patterns consistent with known prognostic relevance.

We then compared protein expression profiles in the virtual CODEX images between different risk groups. The low-risk group was enriched for immune markers such as CD20, CD3e, CD8 and granzyme B. In contrast, the high-risk group showed increased expression of proliferation (Ki-67), epithelial (epithelial cell adhesion molecule (EpCAM) and pan-cytokeratin (PanCK)), stromal (alpha-smooth muscle actin) and angiogenesis (CD31) markers (Fig. 5c; all $P < 0.001$). Comparisons across all 40 biomarkers are provided in Supplementary Fig. 17.

To further explore spatial co-localization patterns of different biomarkers, we quantified the prevalence of image tiles with high co-expression values for pairs of biomarkers (Methods). We observed that the fraction of $CD3e^+/PanCK^+$ tiles was significantly higher in the low-risk group, consistent with intraepithelial immune infiltration. Conversely, the high-risk group exhibited significantly higher fractions of $Ki-67^+/PanCK^+$ and $CD44^+/EpCAM^+$ tiles, reflecting a proliferative tumor and cancer stem cell niche ($P < 0.001$ for all comparisons; Fig. 5d).

To confirm the prognostic significance of these spatial patterns, we computed the number of image tiles with HEX-predicted high co-expression patterns at the patient level and stratified patients into high- and low-count groups. In the NLST cohort, patients with high prevalence of $CD3e^+/PanCK^+$ tiles had better survival outcomes ($HR = 0.58$; $P = 0.001$), whereas those with high prevalence of $Ki-67^+/PanCK^+$ or $CD44^+/EpCAM^+$ tiles had significantly worse survival ($HR = 1.64$ ($P = 0.001$) and $HR = 1.56$ ($P = 0.005$), respectively) (Fig. 5e). To further validate these findings, we computed the same spatial biomarkers using the actual CODEX images in the Stanford-TMA cohort and, indeed, the same prognostic patterns were observed for all three biomarkers (Extended Data Fig. 10). Overall, these results demonstrate that our HEX approach effectively integrates tissue morphology with spatially resolved protein expression, enabling biologically interpretable prediction of patient prognosis.

## HEX improves immunotherapy response prediction in advanced lung cancer

Finally, we evaluated the ability of HEX to predict response and outcomes after ICIs based on standard H&E-stained WSIs. We collected digitized H&E slides, clinical data and outcomes for 148 patients with advanced NSCLC treated with ICIs against programmed death-1

protein (PD-1) or programmed death ligand 1 (PD-L1). HEX was used to generate virtual CODEX maps from the H&E images, which were then integrated with histology features for outcome prediction in our MICA framework.

For objective response prediction, MICA achieved an area under the receiver operating characteristic curve (AUC) of 0.82 (95% confidence interval (CI): 0.73–0.90), significantly outperforming the H&E-only model with an AUC of 0.72 (95% CI: 0.59–0.80) and the virtual CODEX-only model with an AUC of 0.75 (95% CI: 0.62–0.84) (all $P < 0.001$; Fig. 6a). The strong performance of the CODEX-only model suggests that biological information critical for determining treatment response is being captured by HEX-inferred spatial proteomics. MICA also demonstrated superior performance over clinically approved tests such as PD-L1 expression (AUC = 0.66; 95% CI: 0.54–0.77; $P < 0.05$) and tumor mutation burden (TMB) (AUC = 0.59; 95% CI: 0.37–0.80; $P < 0.05$).

For predicting progression-free survival (PFS), MICA achieved a C-index of 0.72 (95% CI: 0.65–0.76), significantly outperforming both the H&E-only model (C-index: 0.62; 95% CI: 0.56–0.67) and the virtual CODEX-only model (C-index: 0.66; 95% CI: 0.61–0.72 (all $P < 0.001$; Fig. 6b). Notably, MICA also outperformed existing biomarkers such as PD-L1 and TMB in predicting PFS ($P < 0.001$; Supplementary Fig. 18). To assess whether MICA offers additional predictive value, we incorporated standard clinical variables including age, sex, body mass index, central nervous system metastases, smoking status and PD-L1 expression. In multivariate Cox regression analysis, MICA remained the most significant independent predictor of PFS (HR = 1.67; 95% CI: 1.36–2.05; $P < 0.001$).

We further evaluated MICA's ability to stratify patients for PFS. Kaplan–Meier analysis revealed clear separation between high- and low-risk groups (HR = 3.11; 95% CI: 2.11–4.57; $P < 0.001$), with median PFS times of 4.4 and 15.1 months, respectively (Fig. 6c). In contrast, PD-L1 expression alone did not significantly distinguish between patient outcomes (Supplementary Fig. 19). Importantly, MICA maintained robust stratification across clinically relevant subgroups, including PD-L1 expression levels, lines of immunotherapy and ICI monotherapy versus chemo–ICI combination regimens (Supplementary Fig. 20). Taken together, these results confirm the added predictive value of virtual spatial proteomics and show that an effective fusion strategy such as MICA allows more accurate prediction of immunotherapy response and outcomes based on standard histopathology. Further decision impact analyses showed improved reclassification over PD-L1 or TMB, higher sensitivity at 90% specificity, and robust stratification within subsets of various PD-L1 strata and those harboring mutations encoding epidermal growth factor receptor (EGFR) changes (Supplementary Results and Supplementary Figs. 21 and 22).

## Biological interpretation of HEX-based immunotherapy response prediction

To better understand the model's prediction in the context of immunotherapy, we analyzed histopathology-inferred virtual spatial proteomics profiles associated with treatment benefit. Using the same approach as in the prognosis interpretation, we grouped image tiles into high- and low-risk categories based on integrated gradient scores. We then compared protein expression values in the virtual CODEX images between the two risk groups. The low-risk group was significantly enriched for immune-related markers such as CD138, CD3e, CD8 and TCF-1, whereas the high-risk group exhibited elevated expression of markers associated with immunosuppressive or fibrotic components, including CD66b, CD163, FAP and collagen IV (all $P < 0.001$; Supplementary Fig. 23).

To gain deeper insight into the cellular states associated with immunotherapy response, we examined the distribution of marker-defined immune and stromal cell states (Fig. 6d). Rather than

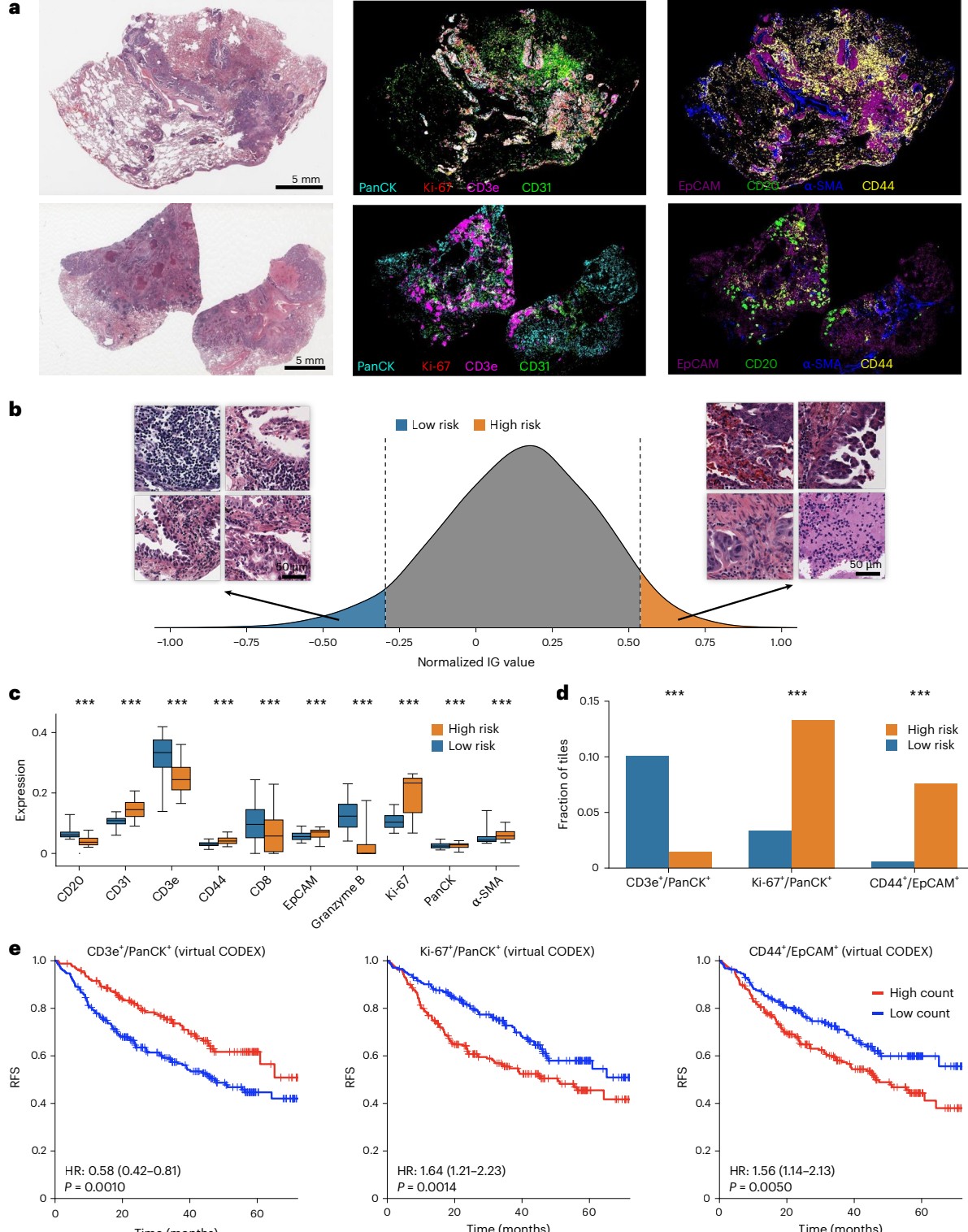

**Fig. 5 | Biological interpretation of MICA-derived risk predictions using HEX virtual spatial proteomics. a**, Representative examples of a high-risk (top) and low-risk (bottom) patient predicted by the MICA model in the NLST cohort. For each patient, original H&E images and corresponding virtual CODEX maps for selected biomarkers generated by HEX are shown. **b**, Histogram of overall integrated gradient (IG) values across all image patches. Integrated gradient values correspond to the predicted risk scores for each image patch. The high- and low-risk groups are defined by the high and low integrated gradient values. **c**, Box plots comparing marker expression distributions between the high- and low-risk groups for selected protein biomarkers ($n$ = 26,519 patches in the high-risk group and $n$ = 26,416 patches in the low-risk group). Central lines indicate medians, boxes span the interquartile range (25th–75th percentiles) and whiskers extend to the 5th–95th percentiles. Group differences were evaluated by two-sided Mann–Whitney $U$-test. **d**, Fractions of image tiles with high co-expression values for three pairs of biomarkers (CD3e[+]/PanCK[+], Ki-67/PanCK[+] and CD44[+]/EpCAM[+]) between the high- and low-risk groups. Two-sided chi-squared tests were used to assess statistical differences between the groups. ***$P$ < 0.001. **e**, Kaplan–Meier analysis based on the number of image tiles with high co-expression patterns for the selected biomarker pairs in **c**. Patients were stratified into high- and low-count groups. Statistical significance was calculated by two-sided log-rank test with no multiple-comparison adjustment.

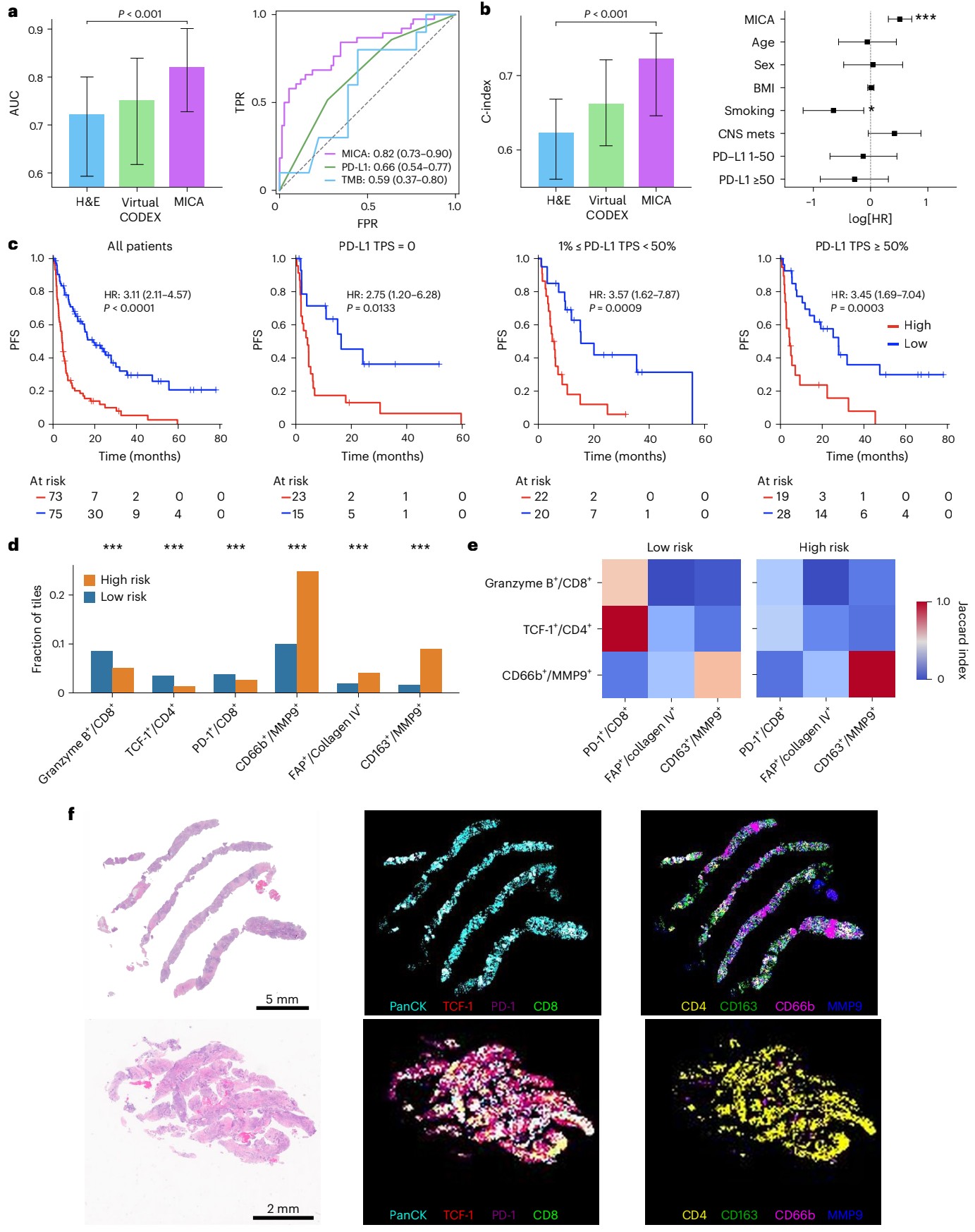

**Fig. 6 | MICA improves immunotherapy response prediction and identifies spatial proteomic signatures in advanced NSCLC. a**, Comparison of model performance for predicting objective response. Left, AUC point estimates for the models based on H&E, virtual CODEX and MICA. Right, ROC curves comparing MICA with PD-L1 expression and TMB. **b**, Comparison of model performance for predicting PFS. Left, C-index point estimates for the models based on H&E, virtual CODEX and MICA. Right, forest plot displaying multivariable Cox regression, including MICA-derived risk scores and clinical covariates. The points denote log[HR] point estimates and the horizontal bars represent 95% CIs. Statistical significance was determined by two-sided Wald test. For the bar graphs in **a** and **b**, the error bars indicate 95% bootstrapped CIs ($n = 1,000$ resamples) derived from aggregated patient-level risk predictions, and statistical significance was assessed using two-sided Wilcoxon signed-rank tests. **c**, Kaplan–Meier analysis of PFS stratified by MICA-predicted risk scores across all patients and PD-L1 expression subgroups (tumor proportion score (TPS) = 0,

$1\% \leq TPS < 50\%$ and $TPS \geq 50\%$). The numbers of patients at risk for each group and time point are given below each graph. Survival differences between high- and low-risk groups were assessed by two-sided log-rank test (cutoff: median). **d**, Fraction of image tiles with marker-defined immune and stromal cell states for selected biomarker pairs (granzyme B$^+$/CD8$^+$, TCF-1$^+$/CD4$^+$, CD66b$^+$/MMP9$^+$, PD-1$^+$/CD8$^+$, FAP$^+$/collagen IV$^+$ and CD163$^+$/MMP9$^+$) between high- and low-risk groups. Two-sided chi-squared tests were used to assess statistical differences (\*\*\*$P < 0.001$). **e**, Normalized Jaccard indices of SCSs. This analysis reveals different spatial co-localization patterns of key cell types for high- and low-risk groups. **f**, Representative examples of a high-risk (top) and a low-risk (bottom) patient predicted by the MICA model. For each patient, original H&E images and corresponding virtual CODEX maps for selected biomarkers generated by HEX are shown. BMI, body mass index; CNS mets, central nervous system metastases; FPR, false positive rate; TPR, true positive rate.

---

exhaustively screening all of the possible combinations, we took a biologically informed approach by focusing on six dual-marker pairs based on their established functional relevance[32–37]. We found that the low-risk group was significantly enriched in granzyme B$^+$/CD8$^+$ (cytotoxic effector T cells), TCF-1$^+$/CD4$^+$ (T helper cells) and PD-1$^+$/CD8$^+$ (exhausted T cells) (all $P < 0.001$). In contrast, the high-risk group was enriched for CD66b$^+$/MMP9$^+$ (pro-tumor neutrophils), FAP$^+$/collagen IV$^+$ (extracellular-matrix-remodeling fibroblasts) and CD163$^+$/MMP9$^+$ (M2-like tumor-associated macrophages), reflecting distinct tumor microenvironments between the two groups.

We further quantified spatial co-localization signatures (SCSs) by assessing the pairwise interaction of cell types and states using a normalized Jaccard index (Fig. 6e). We found that the low-risk group was characterized by coordinated spatial co-localization of various T cell subpopulations, including effector and exhausted T cells and T helper cells. In contrast, the high-risk group exhibited stronger co-localization of immunosuppressive tumor-associated macrophages and neutrophils (CD66b$^+$/MMP9$^+$ and CD163$^+$/MMP9$^+$). To illustrate these concepts, we present two examples with paired H&E and virtual CODEX visualizations for model-predicted high- and low-risk patients, showcasing how the model captures key spatial proteomic features associated with immunotherapy response (Fig. 6f).

Finally, to assess the prognostic relevance of these SCSs, we calculated the number of tiles positive for each SCS per patient and stratified patients into high- and low-count groups (Supplementary Fig. 24). Patients with high counts of CD66b$^+$/MMP9$^+$ and CD163$^+$/MMP9$^+$ SCSs had significantly worse outcomes (HR = 2.38; 95% CI: 1.63–3.48; $P < 0.001$), whereas patients enriched in TCF-1$^+$/CD4$^+$ and PD-1$^+$/CD8$^+$ or granzyme B$^+$/CD8$^+$ and PD-1$^+$/CD8$^+$ signatures showed significantly improved outcomes (HR = 0.58 (95% CI: 0.39–0.85; $P = 0.005$) and HR = 0.60 (95% CI: 0.40–0.88; $P = 0.009$), respectively). However, the expression of single biomarkers did not stratify patients for PFS (Supplementary Fig. 25). These findings confirm the predictive value of spatially co-expressed immune and stromal cell states and demonstrate that distinct cellular interactions—rather than individual biomarkers alone—shape treatment outcomes in advanced NSCLC. This highlights the potential of HEX-generated virtual spatial proteomics for uncovering spatial patterns of the tumor microenvironment linked to therapeutic response and resistance.

### Spatial transcriptomics comparison and scaling analyses

HEX outperformed state-of-the-art spatial transcriptomics predictors (BLEEP[38] and OmiCLIP[39]), and the H&E plus virtual CODEX pipeline (MICA-CODEX) exceeded the H&E plus virtual spatial transcriptomics pipeline (MICA-ST) for objective response and PFS (Supplementary Results, Methods and Supplementary Figs. 26 and 27). Scaling analyses showed monotonic gains with larger training sets and marker panels, with diminishing returns beyond around eight to ten WSIs and around 20 markers (Supplementary Figs. 28 and 29).

## Discussion

In this study, we present an AI model to computationally generate virtual spatial proteomics profiles from standard histopathology slides. Built on a pathology foundation model, our HEX approach predicts spatially resolved protein expression for 40 biomarkers from H&E images with high accuracy. This approach has been extended to 34 tissue types and new protein markers, demonstrating substantial performance gains over alternative methods for protein expression prediction from H&E images. We further develop a multimodal data integration method that combines the original H&E stain- and AI-derived virtual spatial proteomics to enhance outcome prediction. The clinical relevance of HEX was evaluated in six independent cohorts of 2,298 patients with NSCLC, as well as 5,019 patients with 12 other cancer types, demonstrating significant improvements in the prediction of prognosis and response to immune checkpoint blockade.

The most significant implication of our study is that HEX only requires standard H&E histopathology to generate virtual spatial proteomics profiles. Traditional methods rely on staining multiple tissue sections[40] but require extensive specimens and, more importantly, cannot resolve complex tissue phenotypes as they do not provide protein co-localization information. Spatial proteomics allow high-resolution mapping of many proteins on the same tissue section. However, due to the high expense and complexity of the assay, existing spatial proteomics studies are limited to a small number of tissue samples[41–43]. In contrast, HEX enables spatial biomarker discovery and validation on a large scale and can be deployed in broad patient populations.

Through extensive clinical evaluation in more than 7,300 patients across 13 cancer types, we demonstrate that integrating histology with virtual spatial proteomics offers a promising approach for improving treatment response and prognosis prediction. In patients with resected NSCLC, it is important to distinguish patients who are cured by surgery alone from those who are at high risk of recurrence and need adjuvant therapy. Currently, the decision of whether to treat early-stage NSCLC with adjuvant therapy is based on clinicopathologic criteria (that is, tumor size ≥4 cm or lymph node involvement). However, these prognostic factors are rather crude and do not allow for accurate prediction of disease recurrence. HEX improves the accuracy of prognosis prediction by more than 20% compared with clinical risk factors. Therefore, it could be used to refine risk stratification and guide personalized adjuvant therapy in early-stage NSCLC.

ICIs have reshaped the treatment landscape for NSCLC, with durable response in a subset of patients. However, the majority derive little or no clinical benefit, and ICIs are associated with significant toxicity and cost. Identifying those patients most likely to benefit remains a critical unmet need[44]. The HEX approach achieved a 24–39% increase in ICI response prediction accuracy over the standard biomarkers PD-L1 and TMB and could thus inform personalized therapy. Currently, ICI monotherapy is a standard of care in patients with NSCLC and tumor

PD-L1 ≥ 50%, given the likelihood of a strong response. However, our HEX model identified that 40% of patients with PD-L1-high tumors did not experience durable response to ICI. These patients may benefit from concurrent chemotherapy in addition to ICI. For patients with intermediate levels of PD-L1 (>0% and <50%), the HEX model could also help to optimize the treatment decisions on ICI monotherapy versus combination ICI and chemotherapy. For patients with tumor PD-L1 = 0%, there is a moderate benefit of dual checkpoint blockade with anti-cytotoxic T-lymphocyte-associated protein 4 plus anti-PD-1 combined with chemotherapy over standard anti-PD-1 plus chemotherapy[45]. However, the triplet combination therapy has increased toxicity, and the optimal regimen is unclear at the patient level. The HEX model identified that 61% of patients with PD-L1-negative tumors did not benefit from anti-PD-1 ICIs. These patients may benefit from novel combination treatment strategies. Finally, patients with tumors harboring EGFR mutations are generally considered refractory to ICIs and therefore do not receive ICIs as their standard treatment. In our analysis, the HEX model identified that 29% of patients with tumors harboring EGFR mutations who nevertheless derived durable clinical benefit from ICIs. These findings suggest that HEX can provide meaningful improvements in patient stratification, with the potential to enhance treatment selection and improve patient outcomes.

The generation of virtual spatial proteomics has dual benefits: it is not only essential for improved prognostic and predictive performance, but also provides critical biological interpretation of the model predictions. In patients treated with ICIs, we showed that responding tumors were characterized by coordinated spatial co-localization of various T cell subpopulations, including exhausted and effector T cells and T helper cells. The spatial proximity between exhausted T cells and cytotoxic effector T cells may reflect an ongoing phenotypic transition within the tumor microenvironment, which could enable effective reinvigoration of T cell cytotoxicity upon ICI administration. Several lines of evidence, including functional studies, support the biological plausibility of our findings. Previously, it has been shown that PD-1 blockade promotes the differentiation of PD-1+ progenitor CD8+ T cells into functional cytotoxic lymphocytes, thereby improving therapeutic efficacy[46,47]. In contrast, we found that non-responding tumors exhibited spatial co-localization of tumor-associated macrophages and neutrophils. Indeed, these myeloid cell types have been shown to promote tumor progression and angiogenesis, facilitate metastatic dissemination[34,48–51] and contribute to the formation of an immunosuppressive tumor microenvironment that impedes effector T cell infiltration[35,52]. Taken together, these findings support that the cell phenotypes and spatial cell–cell interactions uncovered by HEX are experimentally validated and mechanistically linked to immunotherapy response and clinical outcomes.

Compared with spatial transcriptomics, an alternative spatial profiling technology, spatial proteomics, offers several advantages. First, proteins are more closely related to cellular functions than RNA transcripts. Second, from a practical view, proteins remain more stable in clinical formalin-fixed paraffin-embedded samples, and images are easier to interpret by pathologists. In contrast, spatial transcriptomics allows spatial profiling of the whole transcriptome and provides a more comprehensive picture of tissue architecture. Previous studies have attempted to predict spatial gene expression from H&E images[53–55]. However, the technical performance has been limited, with an average Pearson's $r$ of ~0.2, partly due to noisy and high-dimensional gene expression data[56]. In contrast, our HEX model achieved much more accurate prediction of targeted protein expression, with $r$ values of 0.73–0.79.

One limitation of current spatial proteomics platforms is the reliance on antibodies for immunofluorescence imaging, which may limit the number of potential targets. In addition, the protein biomarker panel may need to be optimized for different cancers or diseases. Emerging technologies such as deep visual proteomics[57] can overcome this challenge by mass cytometry that achieves substantially greater proteome coverage.

The ability to leverage complementary information in a multi-omics framework is crucial as it provides a holistic view of tissue architecture and function[58]. Our computational approach for spatial proteomics–histopathology integration can be extended to other data types, such as spatial transcriptomics and metabolomics. These new molecular insights will further expand our existing knowledge about disease biology[59].

In conclusion, our study provides a low-cost and scalable approach to the study of spatial biology in standard histopathology and enables the discovery and clinical translation of reliable and interpretable biomarkers for precision medicine.

## Online content

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

## Methods

### Study design

This study included a total of 2,308 patients across seven independent cohorts of patients with NSCLC. Three cohorts—Stanford-WSI, Stanford-TMA and TA-TMA—contained same-section co-registered H&E and 40-plex CODEX images and were used for training and validation of the HEX model for AI-enabled generation of spatial proteomics. To evaluate HEX's generalizability and robustness across different tissue types and histological protocols, we conducted external validation using a pan-cancer dataset comprising matched 57-plex CODEX and H&E images from 34 tissue types. To assess the clinical relevance of HEX-derived features, we analyzed H&E-stained slides from six cohorts—NLST, TCGA, PLCO, Stanford-TMA, TA-TMA and Stanford immuno-oncology (Stanford-IO)—through multimodal integration for outcome prediction. Across these cohorts, 2,150 patients had available data on RFS, DSS or overall survival and 148 patients treated with ICIs had objective response and PFS outcomes. To assess the clinical utility beyond NSCLC, we further evaluated HEX for prognosis prediction across 12 additional cancer types in TCGA ($n$ = 5,019). This retrospective study was approved by the Stanford University Institutional Review Board. An overview of the study cohorts and experimental design is provided in Fig. 1 and detailed inclusion and exclusion criteria are provided in Extended Data Fig. 2. Comprehensive descriptions of each patient cohort and associated tumor samples are provided in the sections below.

### CODEX with high-plex PhenoCycler imaging

**Tissue preparation.** Human lung formalin-fixed paraffin-embedded tissue blocks and TMAs were sectioned at a thickness of 5 μm and mounted directly onto SuperFrost Plus microscope slides (Thermo Fisher Scientific). Slides were stored at 4 °C.

**Marker panel design.** We selected 40 protein markers based on their known biological roles and clinical relevance in NSCLC and the tumor microenvironment. These markers include lineage-specific, structural, functional, immune checkpoint, stromal, vascular, extracellular matrix remodeling and antigen presentation proteins. The 40-plex panel was optimized to capture a broad spectrum of tumor–immune phenotypes and spatial interactions at single-cell resolution.

**Antibodies.** Antibody conjugation was performed by the Stanford Cell Sciences Imaging Facility using carrier-free antibodies. Briefly, antibodies were buffer exchanged and reduced using centrifugation filters and a reduction master mix. Barcodes were resuspended and incubated with the reduced antibodies for 2 h at room temperature, followed by purification and storage in antibody stabilization buffer at 4 °C. Commercial antibodies were primarily sourced from Akoya Biosciences and other vendors (Supplementary Table 1). For antibodies not commercially available, custom conjugations were performed using pre-validated primary antibodies and designed barcodes, following the manufacturer's recommended protocol. All antibodies underwent validation for specificity and performance using PhenoCycler multiplexed imaging.

**High-plex staining and imaging.** Tissue staining and imaging were performed following the PhenoCycler-Fusion User Guide version 2.1.0. Briefly, after overnight baking at 60 °C, slides were deparaffinized using a standard histological procedure. Antigen retrieval was conducted in Tris-EDTA buffer (pH 9.0) in a pressure cooker at 120 °C for 20 min under high pressure. After cooling to room temperature, slides were rinsed twice in double-distilled water and incubated in a photobleaching buffer for 45 min. After extensive washing in 1× phosphate-buffered saline, slides were hydrated and stained, per the manufacturer's protocol. Slides were incubated in the conjugated antibody cocktail overnight at 4 °C in a humidity chamber, followed by post-staining fixation and washing steps, as instructed in the manufacturer's user manual.

High-plex immunofluorescence imaging was performed using the PhenoCycler 2.0 platform with a 20× objective. Images were acquired in DAPI, AF488, Atto550 and AF647 channels and automatically processed into a qpTIFF file.

### H&E staining of the same section after CODEX

Following PhenoCycler imaging, slides were stored at 4 °C in the storage buffer provided with the PhenoCycler-Fusion Stain Kit. Flow cells were removed from the slides according to the manufacturer's instructions. After extensive washing with 1× PhenoCycler buffer, slides were stained with standard H&E protocols and imaged on a Leica Aperio AT2 scanner at 40× magnification.

### CODEX and H&E image registration and dataset construction

The DAPI images in CODEX were registered to the hematoxylin channel of the H&E images using PALOM (version 2024.4; https://github.com/labsyspharm/palom), with key parameters including full-resolution alignment (level=0), affine initialization using 4,000 keypoints at pyramid level 1, and block-wise shift refinement. Registered images were divided into non-overlapping tiles with a size of 224 × 224 pixels, and the average expression of each of the 40 protein biomarkers in CODEX was computed for the corresponding H&E image tile. We analyzed three cohorts with matched H&E and 40-plex CODEX staining: (1) Stanford-WSI, comprising ten WSIs from Stanford Hospital, yielded 754,836 tiles annotated with 40-marker expression profiles and was used for model training and cross-validation, with patients split 80/20 into training and test sets in each fold; (2) Stanford-TMA, including 264 TMA cores, contributed 25,383 tiles; and (3) TA-TMA, comprising 108 TMA cores from TissueArray, contributed 39,313 tiles. Stanford-TMA and TA-TMA datasets were used exclusively for independent validation of H&E-based protein expression.

To evaluate the spatial accuracy of CODEX–H&E co-registration, we performed independent cell segmentation on each modality using modality-specific pipelines for Stanford-WSI. The transformation computed by PALOM was then applied to map CODEX cell centroids into the H&E coordinate frame. For each cell in the H&E image, we computed the Euclidean distance to the nearest mapped CODEX cell. We also measured the diameter of each cell nucleus as a biological reference scale. As shown in Supplementary Fig. 30, 99.2% of cell–cell distances between H&E and CODEX were smaller than the median cell nucleus diameter, indicating that accurate cross-modality alignment is achieved with a subcellular resolution.

### Pan-cancer CODEX–H&E dataset for protein prediction

For external validation, we used a pan-cancer dataset comprising 206 TMA cores spanning 34 distinct tissue types, including malignant, benign and normal tissues. The tissue types include: adrenal gland, biliary system, bone, bone marrow, brain, breast, cervix, colon, kidney, liver, lung, lymph node, meninges, muscle, musculoskeletal tissue, nasopharyngeal tissue, nerve, ovary, pancreas, parathyroid, placenta, pleura, prostate, salivary gland, skin, soft tissue, spleen, stomach, tendon, testis, thymus, thyroid, tonsil and uterus[11]. Samples were obtained from the University of Bern, prepared using a distinct H&E staining protocol and digitized at 20× magnification using a Keyence BZ-X710 scanner. CODEX imaging was performed with a 57-marker panel, including 33 protein markers not present in the original NSCLC panel, allowing evaluation of HEX's ability to generalize to novel targets and diverse tissue types.

### Patient cohorts for clinical outcome prediction

**Cohorts for prognosis prediction.** We assembled five NSCLC cohorts totaling 2,150 patients with H&E-stained images and associated clinical outcomes to evaluate the prognostic utility of HEX-generated virtual spatial proteomics. The NLST[28], a large multicenter randomized trial conducted in the USA, served as the training set and included participants with NSCLC and available RFS data. Four independent validation

cohorts were used: (1) the TCGA cohort, comprising patients with lung adenocarcinoma and squamous cell carcinoma whose WSIs and RFS data were downloaded from the Genomic Data Commons; (2) the PLCO Cancer Screening Trial cohort[29], with deidentified WSIs and DSS data obtained via the National Cancer Institute's Cancer Data Access System; (3) the Stanford-TMA cohort; and (4) the TA-TMA cohort, with Stanford-TMA and TA-TMA both comprising NSCLC cases along with curated WSIs and annotated RFS and overall survival outcomes, respectively. All cohorts were restricted to histologically confirmed NSCLC and included clinical metadata such as stage, survival endpoints and demographic variables (Supplementary Tables 2 and 3).

**Cohort for immunotherapy response prediction.** We assessed the clinical utility of HEX-generated virtual spatial proteomics for predicting immunotherapy response and outcomes in the Stanford-IO cohort. The cohort included 148 patients with advanced (metastatic or recurrent) NSCLC who received anti-PD-1 or anti-PD-L1 ICIs at Stanford. The inclusion criteria were treatment with ICIs with or without concurrent chemotherapy and H&E-stained slides available from a pretreatment core needle or surgical biopsy. The best overall response was categorized as responder (complete or partial response) or non-responder (stable or progressive disease). PFS was defined as the time from treatment initiation to clinical or radiographic progression or death. Patients without progression were censored at the date of last follow-up. Detailed clinicopathological characteristics are summarized in Supplementary Table 4.

## WSI processing

All H&E images were processed at 40× magnification (~0.25μm px$^{-1}$). Slides scanned at lower resolutions were rescaled to 40× to ensure consistency across datasets. This resolution is compatible with the MUSK backbone, which was pretrained using multi-scale image augmentations including 10×, 20× and 40× fields of view[22]. To minimize the influence of image artifacts, regions containing pen marks, tissue folds or blurring were identified and excluded using color-based filtering techniques[60]. Before feature extraction, all H&E tiles were standardized using pixel-wise normalization (mean = 0.5; s.d. = 0.5), matching the MUSK training distribution. For CODEX data, each biomarker channel was normalized by scaling the intensities to the [0, 1] range based on the 99th percentile of its global intensity distribution. For each processed H&E image, the HEX model was applied to generate a corresponding virtual spatial proteomics profile, which was subsequently used for multimodal data integration. For orthogonal validation using IHC, we co-registered adjacent-section IHC and H&E WSIs, resampled all slides to 0.25 μm px$^{-1}$ resolution and extracted protein signals from the 3,3'-diaminobenzidine (DAB) channel via hematoxylin–eosin–DAB (HED) color deconvolution.

## HEX model architecture and training strategy

The HEX model was designed to predict the expression level for 40 protein biomarkers simultaneously, given an input H&E image patch (typically 224 × 224 pixels in this study). An overview of the model architecture is visualized in Fig. 1a and Extended Data Fig. 1. The backbone of HEX is built on a pretrained pathology foundation model, such as MUSK, to extract visual features from H&E patches. A two-stage regression head follows: a linear layer reducing the visual embedding to 256, followed by ReLU and dropout, then another linear layer projecting to 128 dimensions with ReLU and dropout and finally a linear output layer producing 40 biomarker predictions.

To improve the predictive robustness and handle the inherent challenges in multiplex imaging data, we integrated two key techniques during fine-tuning: FDS[61] and an ALF[62]. Spatial proteomics data such as CODEX exhibit substantial target imbalance: some biomarkers are ubiquitously expressed, whereas others appear infrequently or in sparse regions. To mitigate this, we adopted FDS, a post-hoc feature calibration technique that reduces the negative impact of data

imbalance by explicitly smoothing features across similar target values. During training, features from the penultimate layer are first collected and stored for each target bin of biomarker $j$ and then updated via the exponential moving average[63]. The calibrated features $\bar{h}$ are obtained by smoothing these bin-level features using a Gaussian kernel $g(\cdot)$:

$$\bar{h} = \bar{C}_b^{\frac{1}{2}} C_b^{-\frac{1}{2}} (h - \mu_b) + \bar{\mu}_b$$

where

$$\mu_b = \frac{1}{N_b - 1} \sum_{i \in b} h_i, C_b = \frac{1}{N_b - 1} \sum_{i \in b} (h_i - \mu_b)(h_i - \mu_b)^T$$

$$\bar{\mu}_b = \sum_{m \in B} g(y_b, y_m) \mu_m, \bar{C}_b = \sum_{m \in B} g(y_b, y_m) C_m$$

$\mu_b$ and $C_b$ are the mean and covariance of the features with each bin $b \in B$, and $N_b$ is the total number of samples in the $b$th bin. FDS was applied directly across all biomarkers to jointly regularize and smooth the feature distributions. This explicit calibration in feature space reduces bias in under-represented targets and stabilizes training in imbalanced regression settings.

To further mitigate the impact of image noise and outliers commonly observed in CODEX data, we incorporated the ALF into the training objective. The ALF generalizes robust loss functions by introducing the learnable shape $\alpha$ and scale $c$ parameters that modulate the tail behavior of the error distribution. This design allows the model to dynamically interpolate between different loss regimens based on data characteristics. When interpreted as the negative log-likelihood of a univariate probability distribution, ALF enables robustness to be automatically adapted during training, improving generalization to noisy or heterogeneous regions in histopathology. The ALF in this study is defined as

$$\mathcal{L}(r, \alpha, c) = \rho(r, \alpha, c) + \log Z(\alpha)$$

where

$$\rho(r, \alpha, c) = \frac{|\alpha - 2|}{\alpha} \left( \left( \frac{(r/c)^2}{\alpha - 2} + 1 \right)^{\alpha/2} - 1 \right)$$

and $Z(\alpha)$ is the partition function, $\alpha$ is the shape parameter, $c$ is the scale parameter and $r$ is the residual error.

## HEX model training details

The network backbone for HEX was pretrained using unified masked modeling (MUSK) on 50 million pathology images from 11,577 patients and one billion pathology-related text tokens. HEX was then trained on matched H&E and spatial proteomic profiles using the Adam optimizer to minimize the ALF, with an initial learning rate of $1 \times 10^{-5}$, exponential decay (factor = 0.95), a batch size of 384 and a dropout rate of 0.5. Training was conducted over 120 epochs: during the first 100 epochs, only the last four encoder layers and regression heads were trained, with the remainder of the backbone frozen; in the final 20 epochs, the regression heads were updated while the entire backbone remained frozen. Training-time data augmentation included random horizontal and vertical flips, rotations, and jittering in brightness, contrast, saturation and hue. The ALF shape ($\alpha$) and scale ($c$) parameters were both initialized to 1. For FDS, the start update parameter and start smooth parameter were set to 0 and 10, respectively.

During inference, HEX scans across the entire tissue using a sliding window with a 224-pixel stride, sequentially generating predictions for each tile and stitching them together into a seamless, fully reconstructed virtual CODEX map. On standard hardware (8× NVIDIA L40S GPUs; batch size: 16), this process takes ~1.3 min per WSI at

40× magnification. HEX also supports single-GPU inference with ≥8 GB of video random access memory. Although these patch and stride sizes served as the default throughout our analyses, the HEX framework supports smaller step sizes, allowing users to generate virtual proteomics maps at higher spatial resolutions as needed. To generate high-resolution virtual CODEX outputs for enhanced visualization of fine-grained biomarker distributions, HEX was retrained on a 1/20 subset of the training dataset using the same input patch size (224 × 224), but with a smaller prediction window (14 × 14 pixels) and a step size of 14 pixels. In this configuration, the model predicts the average expression over the central 14 × 14 region of each input patch, enabling finer spatial resolution in the reconstructed virtual proteomic maps.

### HEX model evaluation and comparison

For a comprehensive evaluation of predictive accuracy, we computed four complementary metrics: Pearson's $r$, which captures the strength of linear relationships between predicted and true expression values; Spearman's $r$, which measures rank-order consistency and reflects the model's ability to preserve relative expression trends; SSIM[64], which quantifies perceptual similarity between spatial maps by considering luminance, contrast and structural patterns; and MSE, which penalizes large deviations in absolute expression levels.

For comparison, we benchmarked HEX against two state-of-the-art generative models for stain translation: a conditional GAN-based method (DeepLIIF)[20] and a contrastive unpaired translation model (Virtual Multiplexer, based on CUT)[18]. These frameworks were selected due to their relevance to histology-to-immunostaining translation tasks. We used their publicly available implementations, adhered to the original training protocols and retrained both models from scratch on our paired H&E–CODEX dataset to predict all 40 biomarkers, ensuring a fair and consistent evaluation. Additionally, we conducted ablation studies demonstrating that both FDS and ALF are critical components for achieving robust and generalizable performance in HEX.

### Model architecture for integrating histology and virtual spatial proteomics

We developed a deep learning framework that integrates H&E histology and virtual spatial proteomics for predicting patient prognosis and immunotherapy response. The MICA model learns how histological features attend to protein expression patterns to make clinically relevant predictions.

MICA begins by extracting tile-level features from H&E images using a pretrained pathology foundation model (MUSK), producing a histology feature bag $H_{bag} = \{h^i\} \in R^{I \times d_k}$, where $I$ is the number of tiles per patient and $d_k$ is the feature dimension. Simultaneously, protein expression features are obtained from each CODEX channel using DINOv2 (ref. 65)—pretrained on 142 million natural images—resulting in a CODEX feature bag $M_{bag} = \{m^j\} \in R^{J \times d_k}$, where $J$ is the number of protein channels.

To fuse these two modalities, we introduce a CODEX-guided co-attention mechanism. This early fusion approach allows the model to selectively focus on histological regions most relevant to the spatial proteomic signals. In the CODEX-guided co-attention layer, we project $M_{bag}$, $H_{bag}$ and $H_{bag}$ into query $Q$, key $K$ and value $V$ representations via three learnable transformations—$P_q$, $P_k$ and $P_v \in R^{d_k \times d_k}$:

$$Q = M_{bag}P_q, \ K = H_{bag}P_k, \ V = H_{bag}P_v$$

The CODEX-guided H&E representation is computed as

$$\widetilde{H}_{coa} = \text{softmax}\left(\frac{QK^T}{\sqrt{d_k}}\right)V$$

This operation enables CODEX-derived features to guide attention over H&E image tiles, highlighting regions of interest and capturing complex multimodal spatial interactions. Finally, two modality-specific multiple-instance learning transformers[66] with global average pooling aggregate features for outcome prediction. Unlike conventional late-fusion approaches, this strategy enables richer cross-modal representation learning and enhances both interpretability and predictive performance.

### Training and evaluation of MICA

For each H&E image, tissue segmentation was performed using the publicly available CLAM repository[67]. Following segmentation, non-overlapping tiles of size 384 × 384 pixels were extracted at 20× magnification from all identified tissue regions. The pretrained MUSK model was then used as a feature extractor to generate histology feature bags. For CODEX, features from each protein channel were extracted using DINOv2 (dinov2_vits14_reg), yielding corresponding CODEX feature bags. Due to variable bag sizes, we employed a mini-batch size of 1, sampling paired H&E and virtual CODEX inputs. No data augmentation was applied for either modality.

For prognosis prediction, MICA was trained using a discrete-time survival loss[68] with the Adam optimizer (learning rate: $1 \times 10^{-5}$; weight decay: $1 \times 10^{-5}$), a batch size of 1 and eight gradient accumulation steps. The model weights of both MUSK and DINOv2 were frozen. No early stopping was applied, and all models were trained for 20 epochs with all parameters trainable.

We evaluated MICA's prognostic performance in three settings: (1) independent NSCLC validation, whereby MICA was trained on the full NLST cohort (RFS) and evaluated on four external NSCLC cohorts (TCGA (RFS), Stanford-TMA (RFS), PLCO (DSS) and TA-TMA (overall survival)); (2) within-cohort robustness, whereby fivefold cross-validation was conducted within the three largest cohorts (NLST, TCGA and PLCO); and (3) pan-cancer generalization, whereby fivefold cross-validation (DSS) across 12 additional TCGA tumor types (5,019 patients) was performed to assess robustness and generalizability.

For immunotherapy response prediction, MICA was trained using either cross-entropy loss (for objective response) or discrete survival loss (for PFS), using the Adam optimizer (learning rate: $2 \times 10^{-4}$; weight decay: $1 \times 10^{-5}$), with a batch size of 1 and eight gradient accumulation steps. Training was performed for 20 epochs, and all models were evaluated using fivefold cross-validation.

For comparative evaluation, we used attention-based multiple-instance learning (AbMIL)[69] as the unimodal baseline for both H&E-only and virtual CODEX-only models. The H&E-only baseline was implemented using image features extracted from the MUSK foundation model, corresponding to the MUSK image approach described in our previous work. The virtual CODEX-only model used DINOv2-extracted features from virtual proteomic images. Additionally, we benchmarked MICA against PORPOISE[30], a deep learning method for late fusion of multimodal data. All models were trained under the same settings as MICA. Performance was evaluated using the C-index, whereas Kaplan–Meier analysis was used to assess the risk stratification capability of each method.

### Biological interpretation of outcome predictions

To explore the biological relevance of the model predictions, we used integrated gradients[31]—a gradient-based attribution method that quantifies the contribution of input features to the model's output. In our context, positive attributions indicate features associated with increased predicted risk, whereas negative attributions correspond to decreased risk. The integrated gradient was computed using the IntegratedGradients function from the Captum library (version 0.4.0), with a zero vector used as the baseline. After calculating integrated gradient values for each H&E tile, we applied min–max normalization across the WSI to facilitate comparison.

We aggregated integrated gradient scores across the entire dataset and defined high- and low-risk groups as the top and bottom 1% of tiles

by integrated gradient value, respectively. For each tile, we computed the average expression of each biomarker in the corresponding CODEX image, enabling comparison of protein expression profiles between risk groups.

We further defined a tile as expressing a high level of a biomarker if its value exceeded the 80th percentile of that marker's distribution across the dataset. Based on this, we assessed the fraction of tiles showing co-expression of biomarker pairs. To characterize SCSs, we counted the number of tiles containing both marker-defined cell states in a given pair and computed a normalized Jaccard index to quantify co-localization patterns in the high- and low-risk groups. Specifically, each of the six pre-specified combinations paired a canonical lineage marker with a functional marker to assess distinct cell states relevant to immunotherapy response (granzyme $B^+$/$CD8^+$, TCF-1$^+$/$CD4^+$, PD-1$^+$/$CD8^+$, CD66b$^+$/MMP9$^+$, FAP$^+$/collagen IV$^+$ and CD163$^+$/MMP9$^+$). Given this limited set of biologically informed hypotheses ($n = 6$), nominal $P$ values were reported without correction for multiple hypothesis testing.

## Statistical analysis

For fivefold cross-validation of protein expression prediction, we reported the mean and standard deviation across all folds. For independent validation, we used non-parametric bootstrapping (1,000 replicates) to estimate 95% CIs for all performance metrics. Kaplan–Meier analysis was used to evaluate the association between MICA-predicted risk scores and patient survival across cohorts. Patients were stratified into high- and low-risk groups based on either the median or an optimized cutoff value (as specified in the figure captions), and survival differences were assessed using the log-rank test. $P < 0.05$ was considered statistically significant. Prognostic model performance was quantified using the C-index. For independent validation, 95% CIs were derived via bootstrapping (1,000 replicates) on all predicted risk scores. For PFS, we reported cross-validated C-index values as the mean across five folds. CIs in cross-validation were estimated by bootstrapping the out-of-sample predictions across validation folds. For immunotherapy response prediction, we used the AUC as the performance metric. Statistical significance between models was assessed using a two-sided Wilcoxon signed-rank test.

## Reporting summary

Further information on research design is available in the Nature Portfolio Reporting Summary linked to this article.

## Data availability

Histopathology and clinical data from the TCGA cohort used in this study are publicly available via the National Cancer Institute Genomic Data Commons portal (https://portal.gdc.cancer.gov) and cBioPortal (https://www.cbioportal.org). WSIs from the NLST cohort can be accessed through The Cancer Imaging Archive (https://wiki.cancer-imagingarchive.net). Requests for access to PLCO data should be submitted through the Cancer Data Access System (https://cdas.cancer.gov/learn/plco). In-house datasets, used with institutional approval, contain sensitive patient information and are not publicly available. Access requests may be directed to the corresponding author and require a brief research proposal and execution of a data use agreement; requests will be reviewed and processed within 2 weeks. Data are made available exclusively for non-commercial academic research. Source data are provided with this paper.

## Code availability

An open-source version of the code used in this study is available at https://github.com/lilab-stanford/HEX. The PALOM toolkit (https://github.com/labsyspharm/palom) was used for co-registration of CODEX and H&E-stained images. Clinical outcome analyses were conducted using the lifelines library (https://github.com/CamDavidsonPilon/lifelines).

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

## Acknowledgements

This study was supported in part by the National Institutes of Health under research grant R01CA290715 from the National Cancer Institute.

## Author contributions

Z.L. and R.L. conceived of and designed the study. R.L., Yuchen Li, F.E., Yuanyuan Li, J.M., X.Z., C.B., T.K., F.M.O., S.W., J.J.N., J.N., R.W. and M.D. acquired private data. Z.L., J.X., X.W., S.Y., Y.C., X.L. and R.L. acquired public data. Z.L. performed the statistical analyses. Z.L. and Yuchen Li trained and validated the deep learning models. R.L., Z.L. and Yuchen Li performed quality control of the data and computational algorithms and drafted and revised the paper. All authors contributed to paper preparation and review.

## Competing interests

The authors declare no competing interests.

## Additional information

**Extended data** is available for this paper at https://doi.org/10.1038/s41591-025-04060-4.

**Correspondence and requests for materials** should be addressed to Ruijiang Li.

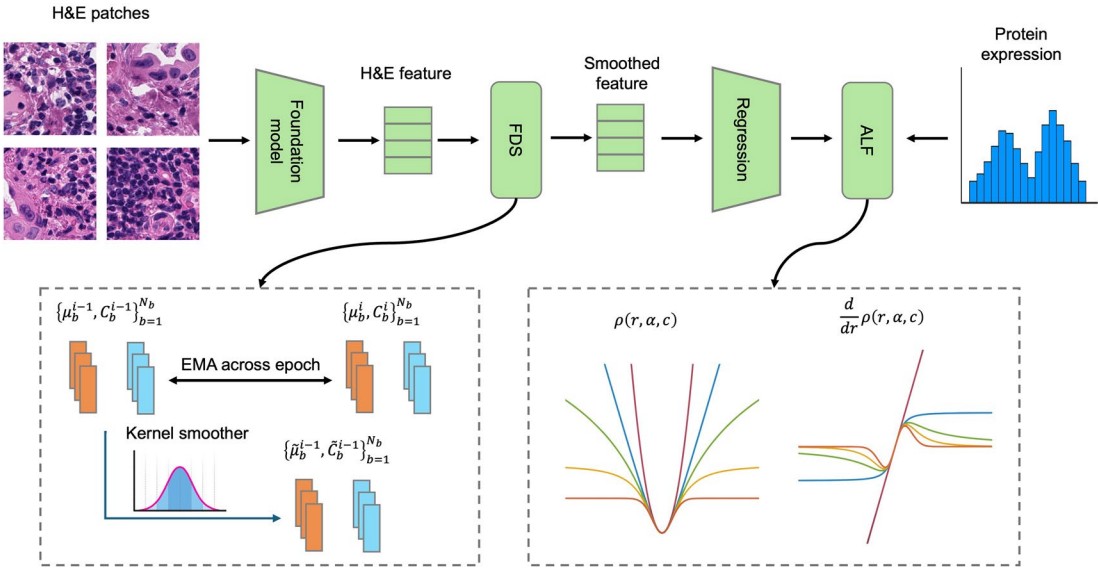

**Extended Data Fig. 1 | Overview of HEX.** HEX is built on a pre-trained pathology foundation model (for example, MUSK) to extract visual features from H&E image patches. A three-layer regression head maps visual embeddings to 40 biomarker predictions via intermediate 256- and 128-dimensional representations with ReLU activations and dropout regularization. To enhance predictive robustness, the model incorporates Feature Distribution Smoothing (FDS) and an Adaptive Loss Function (ALF) during training to address the challenges of multiplexed imaging data. EMA, exponential moving average.

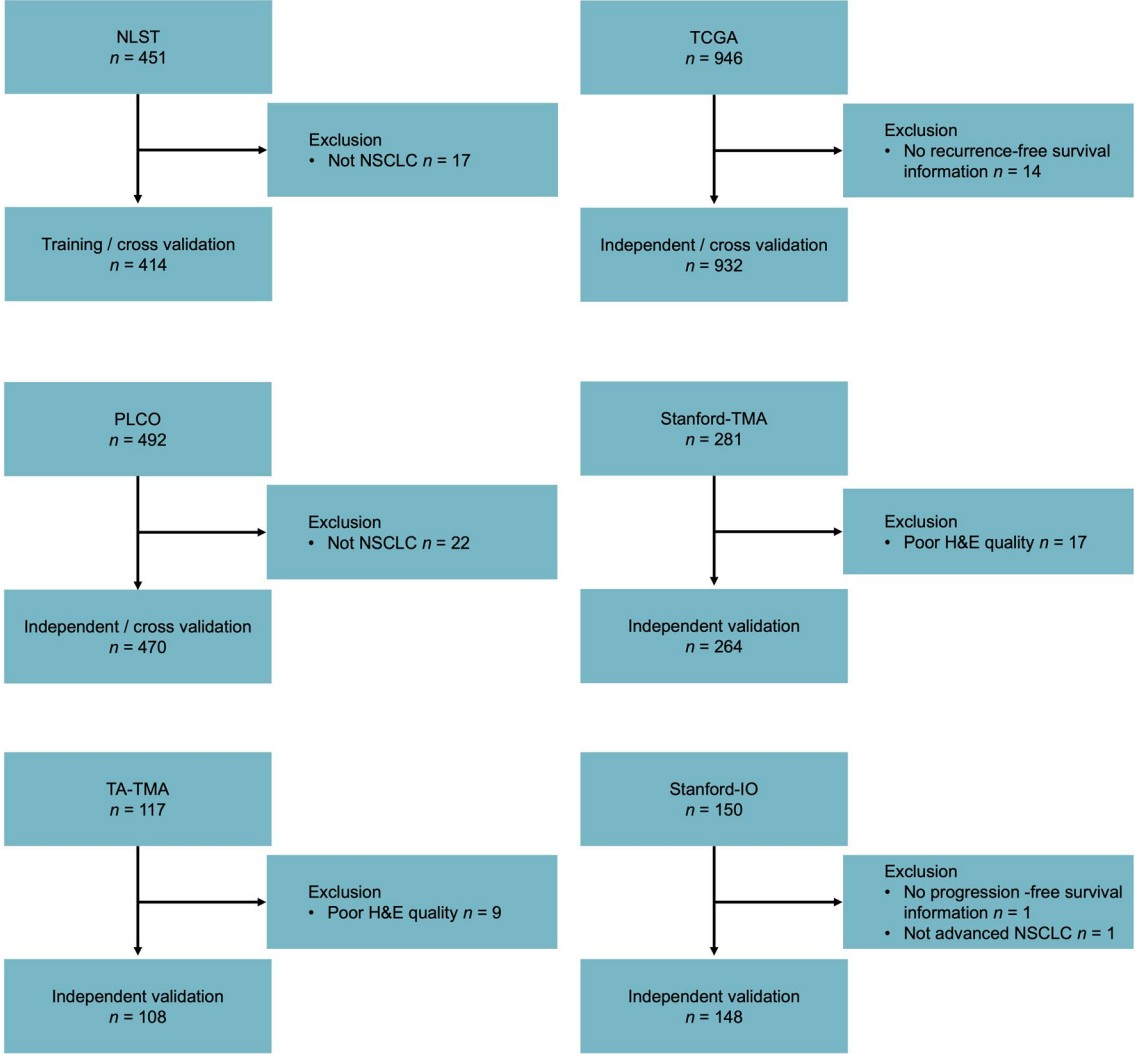

**Extended Data Fig. 2 | Sample inclusion and exclusion across six cohorts.** Flow chart depicting patient/sample inclusion and exclusion for the six study cohorts—NLST, TCGA, PLCO, Stanford-TMA, TissueArray TMA (TA-TMA), and the Stanford immuno-oncology (Stanford-IO) cohort.

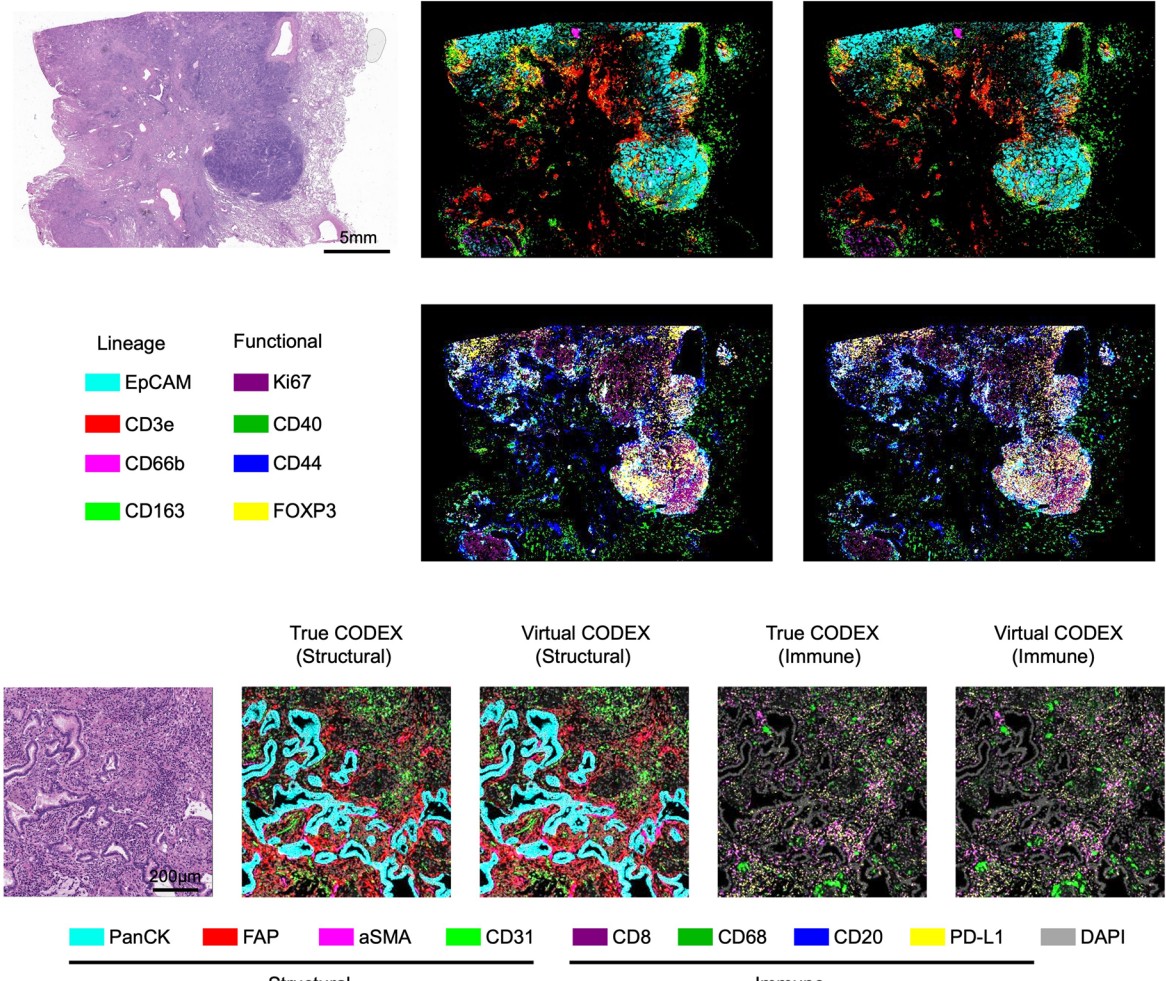

**Extended Data Fig. 3 | Additional representative visualizations of HEX-predicted virtual spatial proteomics profiles.** The virtual CODEX images demonstrate high resemblance to the true images across structural, immune, lineage, and functional biomarkers at both standard and fine spatial resolutions, with a 224- and 14-pixel patch sizes, respectively. Similar results were obtained in 10 independent WSIs.

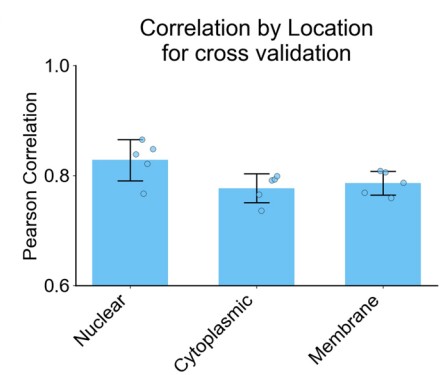

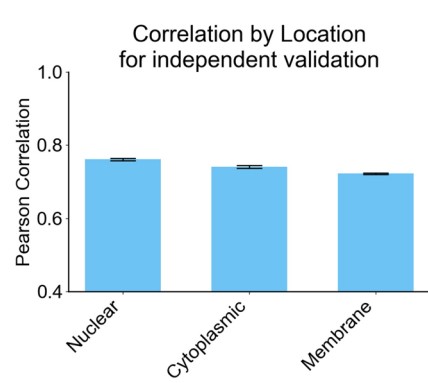

**Extended Data Fig. 4 | HEX protein prediction accuracy by subcellular localization.** Average Pearson correlations for nuclear, cytoplasmic, and membrane markers in cross-validation (**a**) and independent validation (**b**). In a, bars represent the mean across five-fold cross-validation on the Stanford-WSI dataset (n = 10 WSIs); dots show individual folds and error bars indicate standard deviation. In b, bars represent point estimates and error bars indicate 95% bootstrap CIs (n = 1,000 resamples) from independent validation cohorts (Stanford-TMA n = 264 TMA cores; TA-TMA n = 108 TMA cores).

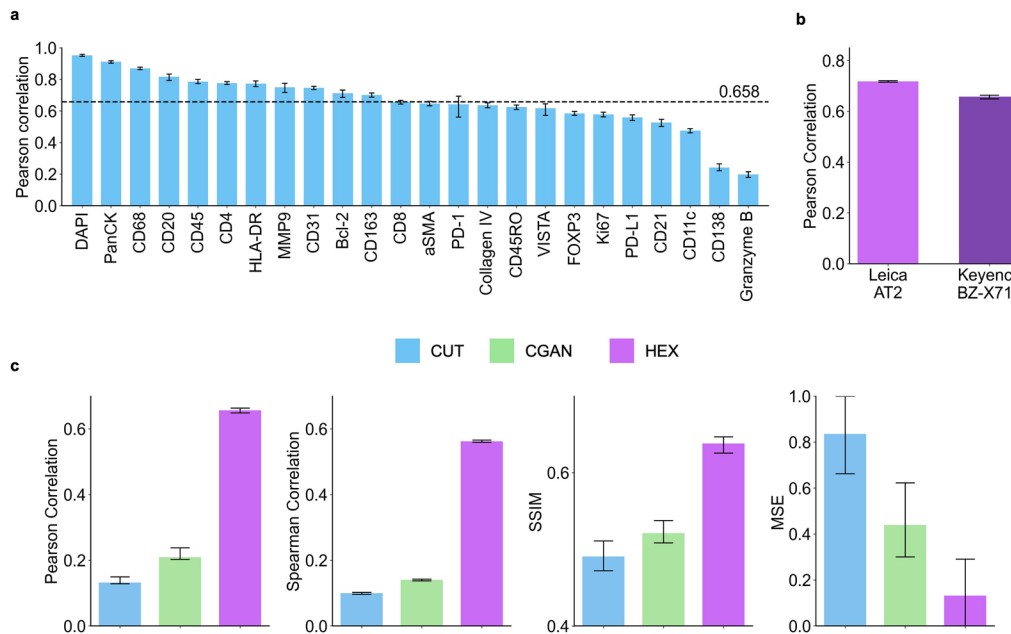

**Extended Data Fig. 5 | External validation of HEX on the Bern pan-cancer dataset. a**, Pearson correlation between predicted and measured protein expression across 24 overlapping markers (n = 206 TMA cores). **b**, Comparison of HEX performance between two imaging platforms: Leica AT2 (Stanford-TMA, n = 264; TA-TMA, n = 108) and Keyence BZ-X710 (Bern, n = 206). **c**, Comparison of HEX with baseline models on the Bern dataset. In **a**–**c**, bars represent point estimates and error bars indicate 95% bootstrap CIs (n = 1,000 resamples).

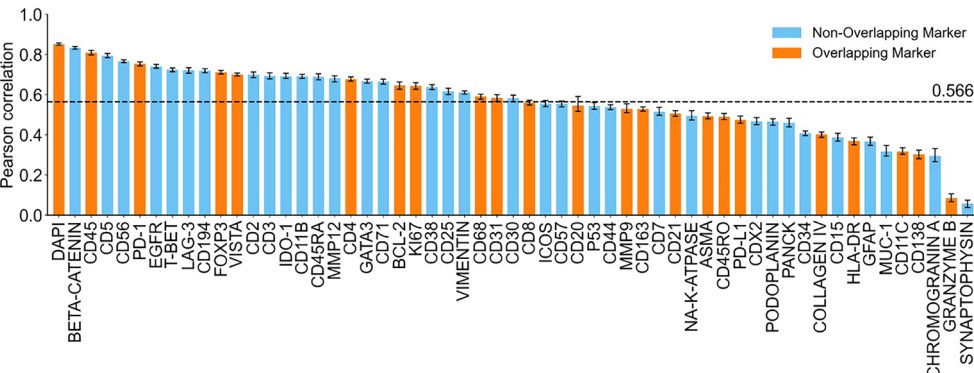

**Extended Data Fig. 6 | HEX retrained on colorectal cancer.** Pearson correlation for 57 markers after retraining HEX on CRC (n = 140 TMA cores). Orange bars denote markers overlapping with the NSCLC panel; blue bars are CRC-specific. Bars represent point estimates and error bars indicate 95% bootstrap CIs (n = 1,000 resamples).

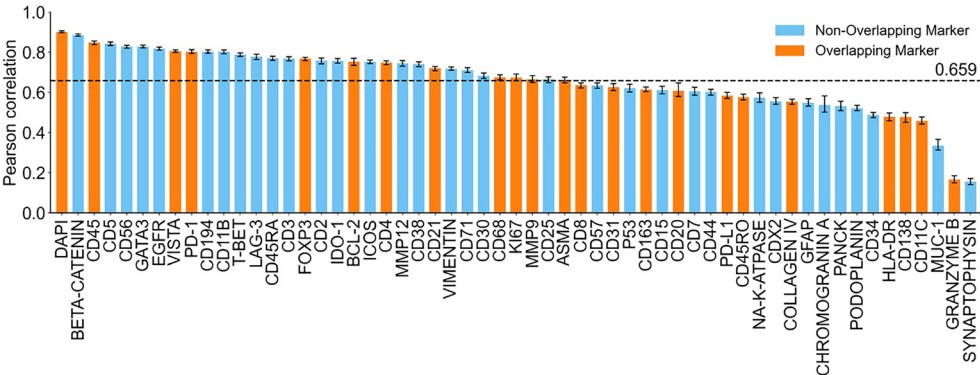

**Extended Data Fig. 7 | HEX fine-tuned on colorectal cancer.** Pearson correlation for 57 markers after fine-tuning the NSCLC-trained HEX model on CRC (n = 140 TMA cores). Orange bars indicate markers overlapping with the NSCLC panel; blue bars are CRC-specific. Bars represent point estimates and error bars indicate 95% bootstrap CIs (n = 1,000 resamples).

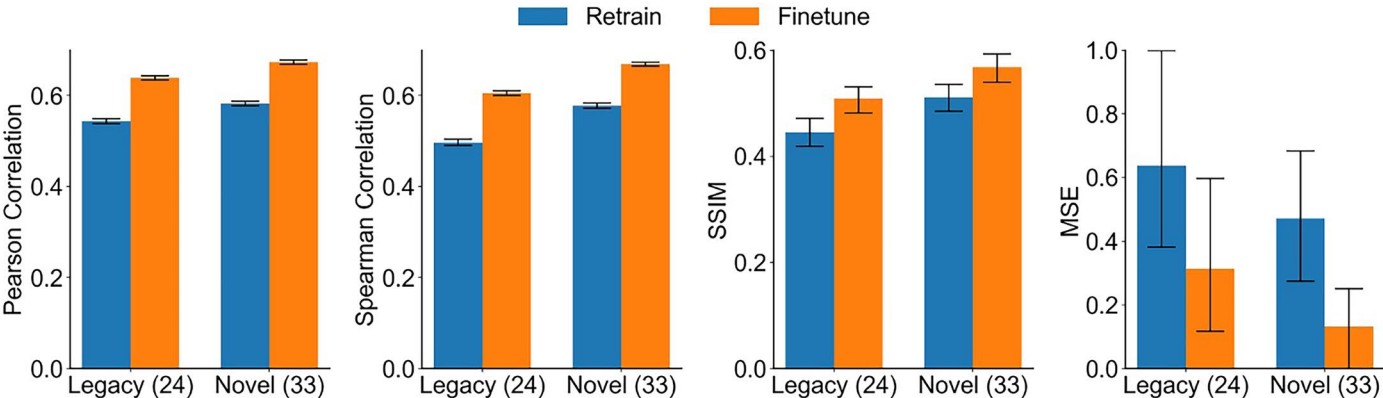

**Extended Data Fig. 8 | Fine-tuning improves prediction across legacy and novel markers.** Performance comparison on legacy (24) and novel (33) markers after retraining or fine-tuning (n = 140 TMA cores). Bars represent point estimates and error bars indicate 95% bootstrap CIs (n = 1,000 resamples).

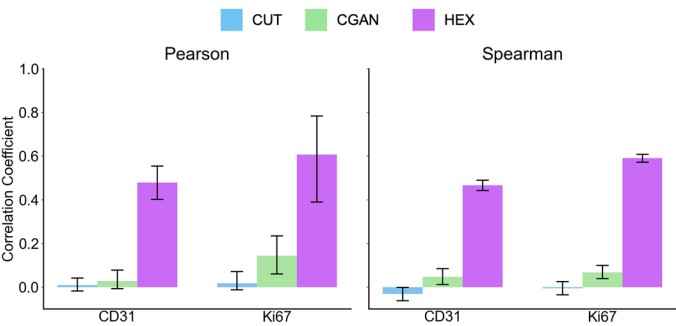

**Extended Data Fig. 9 | Orthogonal validation of HEX-derived protein expression using IHC.** Comparison of HEX with two generative AI models using Pearson (left) and Spearman (right) correlation coefficients for CD31 and Ki67 across three lung samples from the ANHIR dataset. Bars represent point estimates and error bars indicate 95% bootstrap CIs (n = 1,000 resamples).

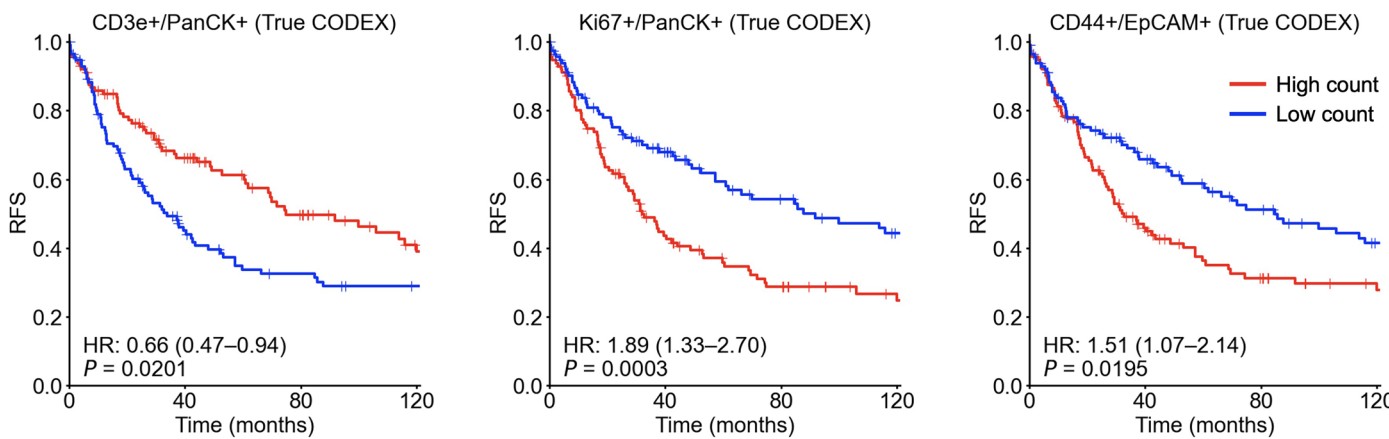

**Extended Data Fig. 10 | Kaplan–Meier analysis based on the number of image tiles with high co-expression patterns for selected biomarker pairs in Fig. 5d in Stanford-TMA cohort.** P values calculated by two-sided log-rank tests. No multiple-comparison adjustment.

# Reporting Summary

## Statistics

For all statistical analyses, confirm that the following items are present in the figure legend, table legend, main text, or Methods section.

| n/a | Confirmed | |
|---|---|---|
| ☐ | ☒ | The exact sample size (*n*) for each experimental group/condition, given as a discrete number and unit of measurement |
| ☐ | ☒ | A statement on whether measurements were taken from distinct samples or whether the same sample was measured repeatedly |
| ☐ | ☒ | The statistical test(s) used AND whether they are one- or two-sided *Only common tests should be described solely by name; describe more complex techniques in the Methods section.* |
| ☐ | ☒ | A description of all covariates tested |
| ☐ | ☒ | A description of any assumptions or corrections, such as tests of normality and adjustment for multiple comparisons |
| ☐ | ☒ | A full description of the statistical parameters including central tendency (e.g. means) or other basic estimates (e.g. regression coefficient) AND variation (e.g. standard deviation) or associated estimates of uncertainty (e.g. confidence intervals) |
| ☐ | ☒ | For null hypothesis testing, the test statistic (e.g. *F*, *t*, *r*) with confidence intervals, effect sizes, degrees of freedom and *P* value noted *Give P values as exact values whenever suitable.* |
| ☒ | ☐ | For Bayesian analysis, information on the choice of priors and Markov chain Monte Carlo settings |
| ☐ | ☒ | For hierarchical and complex designs, identification of the appropriate level for tests and full reporting of outcomes |
| ☒ | ☐ | Estimates of effect sizes (e.g. Cohen's *d*, Pearson's *r*), indicating how they were calculated |

*Our web collection on statistics for biologists contains articles on many of the points above.*

## Software and code

Policy information about availability of computer code

| | |
|---|---|
| Data collection | High-plex image acquisition was performed on the PhenoCycler 2.0 platform per manufacturer protocol. H&E images were obtained post-CODEX on a Leica Aperio AT2 scanner. All deep learning experiments were performed using Python (3.10.15). |
| Data analysis | Data analysis was performed using Python (3.10.15) and PyTorch (2.4.0+cu118), with key libraries including lifelines, scikit-learn, and matplotlib. Analyses utilized open-source toolkits such as MUSK, PALOM, DINOv2, CLAM, MCAT, and imbalanced-regression. Deep learning experiments were run on NVIDIA GPUs (L40S x8) with CUDA 11.8 and cuDNN 9.1 (Ubuntu 22.04). An open-source version of the code is available at https://github.com/lilab-stanford/HEX. |

For manuscripts utilizing custom algorithms or software that are central to the research but not yet described in published literature, software must be made available to editors and reviewers. We strongly encourage code deposition in a community repository (e.g. GitHub). See the Nature Portfolio guidelines for submitting code & software for further information.

## Data

Policy information about availability of data

All manuscripts must include a data availability statement. This statement should provide the following information, where applicable:

- Accession codes, unique identifiers, or web links for publicly available datasets
- A description of any restrictions on data availability
- For clinical datasets or third party data, please ensure that the statement adheres to our policy

The TCGA clinical and histopathology data analyzed in this study are publicly accessible via the NCI Genomic Data Commons (https://portal.gdc.cancer.gov) and cBioPortal (https://www.cbioportal.org). NLST whole-slide image (WSI) data are publicly available from The Cancer Imaging Archive (https://wiki.cancerimagingarchive.net). PLCO data can be requested through the Cancer Data Access System (https://cdas.cancer.gov/learn/plco). Data from the Stanford-WSI, Stanford-TMA, TA-TMA, and Stanford-IO cohorts are subject to institutional restrictions due to patient confidentiality. Researchers may request access to these datasets from the corresponding author by submitting a brief research proposal and completing a data use agreement, with usage restricted to academic, non-commercial research purposes.

## Research involving human participants, their data, or biological material

Policy information about studies with human participants or human data. See also policy information about sex, gender (identity/presentation), and sexual orientation and race, ethnicity and racism.

| | |
|---|---|
| Reporting on sex and gender | Not applicable. |
| Reporting on race, ethnicity, or other socially relevant groupings | Socially relevant characteristics of the study participants (when available) are provided in Supplementary Table 2, 3, and 4. |
| Population characteristics | Detailed population characteristics for the in-house datasets are provided in Supplementary Table 2, 3, and 4. |
| Recruitment | No patient recruitment was necessary for the use of whole-slide images retrospectively. |
| Ethics oversight | This retrospective study was approved by the institutional review board at Stanford University. |

Note that full information on the approval of the study protocol must also be provided in the manuscript.

# Field-specific reporting

Please select the one below that is the best fit for your research. If you are not sure, read the appropriate sections before making your selection.

☒ Life sciences  ☐ Behavioural & social sciences  ☐ Ecological, evolutionary & environmental sciences

For a reference copy of the document with all sections, see nature.com/documents/nr-reporting-summary-flat.pdf

# Life sciences study design

All studies must disclose on these points even when the disclosure is negative.

| | |
|---|---|
| Sample size | A total of 2,298 patients with histologically confirmed non-small cell lung cancer (NSCLC) were included across seven independent cohorts. For prognosis prediction, five cohorts comprising 2,150 patients with available H&E-stained whole-slide images (WSIs) and clinical outcome data were used: the National Lung Screening Trial (NLST), The Cancer Genome Atlas (TCGA), the Prostate, Lung, Colorectal, and Ovarian Cancer Screening Trial (PLCO), and two tissue microarray cohorts—Stanford-TMA and TA-TMA—used for external validation. For immunotherapy response analysis, the Stanford-IO cohort included 148 patients with advanced NSCLC treated with PD-1 or PD-L1 immune checkpoint inhibitors, with corresponding H&E slides from pretreatment biopsies. Cohort-specific clinical characteristics and outcome definitions are provided in Supplementary Tables 2–4. |
| Data exclusions | Samples were excluded based on predefined clinical and technical criteria. Specifically, cases were removed if they were not histologically confirmed as NSCLC, had poor-quality H&E slides unsuitable for analysis, or lacked recurrence-free survival or progression-free survival data. One additional case was excluded from the immunotherapy cohort due to not meeting the advanced NSCLC criteria. All exclusions were made prior to model training or evaluation. |
| Replication | Model performance was evaluated using both cross-validation and independent validation. For cross-validation, models were trained and tested using five-fold splits, and all experiments were repeated five times to assess reproducibility. For independent validation, models trained on one cohort were evaluated on external datasets without retraining. All replication attempts were successful and showed consistent results across runs and datasets. |
| Randomization | For five-fold cross-validation, patients were randomly assigned to training and validation sets in each fold. |
| Blinding | Blinding was not necessary because the experiments were based on digitized histology slides. |

# Reporting for specific materials, systems and methods

We require information from authors about some types of materials, experimental systems and methods used in many studies. Here, indicate whether each material, system or method listed is relevant to your study. If you are not sure if a list item applies to your research, read the appropriate section before selecting a response.

## Materials & experimental systems

| n/a | Involved in the study |
|-----|----------------------|
| ☐ ☒ | Antibodies |
| ☒ ☐ | Eukaryotic cell lines |
| ☒ ☐ | Palaeontology and archaeology |
| ☒ ☐ | Animals and other organisms |
| ☒ ☐ | Clinical data |
| ☒ ☐ | Dual use research of concern |
| ☒ ☐ | Plants |

## Methods

| n/a | Involved in the study |
|-----|----------------------|
| ☒ ☐ | ChIP-seq |
| ☒ ☐ | Flow cytometry |
| ☒ ☐ | MRI-based neuroimaging |

## Antibodies

| Antibodies used | Detailed in Methods. |
|-----------------|----------------------|
| Validation | All antibodies used are commercially available and have been validated by the corresponding vendors |

## Plants

| Seed stocks | *Report on the source of all seed stocks or other plant material used. If applicable, state the seed stock centre and catalogue number. If plant specimens were collected from the field, describe the collection location, date and sampling procedures.* |
|-------------|----------------------|
| Novel plant genotypes | *Describe the methods by which all novel plant genotypes were produced. This includes those generated by transgenic approaches, gene editing, chemical/radiation-based mutagenesis and hybridization. For transgenic lines, describe the transformation method, the number of independent lines analyzed and the generation upon which experiments were performed. For gene-edited lines, describe the editor used, the endogenous sequence targeted for editing, the targeting guide RNA sequence (if applicable) and how the editor was applied.* |
| Authentication | *Describe any authentication procedures for each seed stock used or novel genotype generated. Describe any experiments used to assess the effect of a mutation and, where applicable, how potential secondary effects (e.g. second site T-DNA insertions, mosiacism, off-target gene editing) were examined.* |

