## [Peer Review File · Nature Medicine]

AI-enabled virtual spatial proteomics from histopathology for interpretable biomarker discovery in lung cancer

Corresponding Author: Dr Ruijiang Li

A version of this paper was originally rejected for publication by Nature Medicine, however that decision was reconsidered after appeal by the authors.

Version 0:

Reviewer comments:

Reviewer #1

(Remarks to the Author)

The authors set out to create an AI model (HEX) capable of spatially resolved protein expression prediction from paired H&E images by leveraging their previous foundation model for H&E expression, MUSK. They use their own cohort of 10 patients' samples with NSCLC using the PCF v2 system. It is important to note that the current model does not contain the authors' pre-trained model head, which would be needed for validation. They evaluate 40 proteins, and independently validate it on TMA and Tissue Array (TA). A subsequent Transformer NN leverages H&E-based and CODEX embeddings (HEX into DINOv2), is then utilized for finetuning on downstream tasks. Additionally, they utilize Integrated Gradients (IG) to identify tiles of high predictive value to downstream clinical characteristics. They compare different foundation model backbones as well as conduct ablation studies to underscore their architectural choices.

Major revisions - It is important that further analyses be carried to elucidate some key points:

1. Testing the model on 10 patients only does not necessarily guarantee the robustness of the metrics. Can the authors test expanding or subsetting their dataset has effects on their metrics (PCC, MSE, etc.) in order to establish scaling laws? Similar scaling should be derived for the relationships of the number of markers and downstream clinically relevant results. It is important to also test bench protocol differences, such as the sensitivity of the model to separate H&E staining protocols and scanners, which often differ between institutions.
2. In the same vein, the authors should validate this on a similarly designed cohort from another tissue, retraining their model to demonstrate generalizability. This need not be newly generated data but can be done using public consortia data too, such as HuBMAP (healthy intestine) and HTAN (BrCa).
3. Given that this model is trained on NSCLC, is the finetuning of this model possible to other tissues, given a similar cohort?
4. Given that this model is trained on 40 specific markers, is this model amenable to a different set of markers, supposing the head is changed?
5. How much better is the predictive accuracy of HEX to just using MUSK (or another foundation model) – e.g. in the MUSK image + report or MUSK image cases from the authors' previous publication?
6. Similarly, there should be a head to head comparison of c-indices that includes the real CODEX data, where it is available (Stanford-TMA and TA-TMA). In every head to head comparison, proper statistical testing should be performed to assess if a difference in performance between any two methods is statistically significant.
7. The authors should include H-Optimus-v1 (or v0) to the comparisons for backbones, as it is the most popular model on HuggingFace and widely used by the community.
8. In the SCS analysis, how were those specific two marker signatures chosen? If all of them were tested, are the provided p-values for those specific ones adjusted for multiple hypothesis testing?

Minor Revisions - Additionally, there are some minor concerns that the authors should clarify:

1. PCF scanned at 40x magnification, but MUSK has been trained at 20x magnification (as have most foundation model), is this something that is addressed through PALOM registration?
2. Are there any trends that relate to protein localization? Nuclear, cytoplasmic, membrane and correlations?
3. The published version of MUSK uses 384x384 patches, but here 224x224 are listed. Was the model reconfigured for this task or just the tiles resized prior to going through the model?
4. Were the H&Es normalized in any way? Similarly for the CODEX data.

5. The Mann-Whitney U test in Extended Data Fig. 22 might be skewed by the large number of patches. It would be great to also note effect sizes for markers there or test against a specific effect size or report similar metrics like Cohen's d.
6. Regarding comparisons with VirtualMultiplexer and DeepLIF, were they pretrained to predict the same 40 markers? Or used only the intersecting ones with this dataset?
7. Some notes should be made on inference time and compute resource requirements. Some of these are available in the GitHub but should also be noted in the Supplementary.
8. Which DINO model was used? (It seems it is dinov2_vits14_reg but this should be mentioned in the text).

(Remarks on code availability)

Reviewer #3

(Remarks to the Author)

Summary of the key results

Spatial proteomics is a relatively novel technology that allows the detection of multiple proteins on a tissue section. Authors have developed an AI model designated as HEX (H&E to protein eXpression), that is able to generate spatial proteomics profiles from standard H&E slides. They selected a total of 40 targets that include immune, stromal and tumor cells. A multimodal data integration approach that combines the H&E and the AI-derived spatial proteomics was shown to enhance outcome prediction. Authors conclude that HEX is a low-cost approach that provides interpretable biomarkers for precision medicine.

Originality and significance: if not novel, please include reference

Other studies have developed AI models able to predict cell types or proteins from HE examples: PMID: 40097393, PMID: 32747659

Data & methodology: validity of approach, quality of data, quality of presentation

The paper is well written and straightforward

Appropriate use of statistics and treatment of uncertainties

Methods appear solid.

Conclusions: robustness, validity, reliability

-The biological interpretation of the model remain superficial, as the study is purely descriptive. To be further explored functional experiments should be further conducted.

Suggested improvements: experiments, data for possible revision

-The sentence in the abstract "rained on 755,000 histopathology images with matched protein expression" as the number of different samples is much less than 755000

-The first part of the results (first paragraph-study overview) is not really needed

-A significant proportion of the c indexes obtained are below 0.8, which limits the use of such models in practice

-Given that the vast majority of the markers used are not lung-specific, investigation in other cohorts of patients with cancers from other organs would be interesting

Clarity and context: lucidity of abstract/summary, appropriateness of abstract, introduction and conclusions

The manuscript is clear and straightforward.

(Remarks on code availability)

Reviewer #4

(Remarks to the Author)

The core breakthrough of the article is the first realization of predicting the spatial expression of 40 proteins from a single H&E section, which solves the key bottlenecks of spatialomics for clinical translation - cost and accessibility. The proposed MICA multimodal fusion framework is also a novel solution. However, the differences with spatial transcriptome prediction efforts could be discussed in more depth, with existing comparisons only briefly mentioning performance differences. No analysis shows how HEX-MICA changes treatment decisions (e.g., % of patients reclassified vs. standard care) or improves patient outcomes. Without evidence of clinical utility beyond statistical metrics, the translational impact remains speculative. Here are the issues

1. The training set (755K image blocks) and validation set (372 samples) were adequately sized, and the multicenter cohort design (6 cohorts/2,298 patients) significantly enhanced conclusion reliability. However, specific margins of error for H&E-CODEX alignment and quality control processes (e.g., PALOM tool parameters mentioned in Page 15) need to be described.

2. Existing results should be compared with spatial transcriptome prediction studies from other studies to highlight protein stability and clinical applicability.

3. Why did the authors choose antibodies to the above 40 proteins, and are they specific enough for tumor marker.
4. In the section on HEX improves immunotherapy response prediction, the authors only state that the system can predict a patient's immune sensitivity, but do not further explore how patient survival can be improved by this system.
5. Test HEX on non-NSCLC samples (e.g., CRC/breast cancer) in supplementary analyses to assess pan-cancer utility.
6. "Virtual proteomics" are never validated with orthogonal methods (e.g., IHC, CyTOF). The study relies entirely on predicted protein patterns to derive biological insights (e.g., Fig 5c-d, Fig 6d-e). Correlations may reflect algorithmic artifacts rather than true biology.

(Remarks on code availability)

Version 1:

Reviewer comments:

Reviewer #1

(Remarks to the Author)

The authors have adequately addressed my comments.

(Remarks on code availability)

N/A

Reviewer #3

(Remarks to the Author)

The different additional experiments have significantly improved the overall manuscript.

(Remarks on code availability)

Reviewer #4

(Remarks to the Author)

The author has answered all my queries and I have no more questions.

(Remarks on code availability)

Response to Comments

We are grateful to all three reviewers to their constructive feedback and valuable suggestions. Below is a summary of the changes made in response to the reviewers' comments (page index is provided for convenience). We have revised our manuscript accordingly, which we believe has been much improved after revision.

Editorial comments:

The required new experiments and data include, but are not limited to, providing evidence that the framework can be applied across cancer types, with additional data for training and validation of the model.

We provide a high-level summary of the key additional data in our revision.

1. **Generalizability across cancer types and tissue protocols.** We have performed extensive evaluation of the HEX model on additional tissue types and new protein markers. We validated our framework on an external pan-cancer dataset of matched CODEX and H&E images for **206 samples from 34 tissue types**, obtained using a different staining protocol and scanner. HEX showed strong protein prediction performance in zero-shot learning, and with fine-tuning can predict 57 proteins markers—including 33 novel markers not present in the original panel. These new results demonstrate HEX's generalizability across diverse tissues and robustness to staining protocols.
2. **Clinical utility across cancer types.** We have validated the prognostic utility of the proposed HEX-MICA multi-modal fusion model on whole-slide histopathology images **across 12 additional cancer types in 5,019 patients**, including breast, colorectal, liver, pancreas, bladder, kidney cancers, etc. The HEX-derived virtual spatial proteomics integrated with H&E via our MICA framework, consistently outperformed both H&E-only and virtual CODEX-only models in prognosis prediction across cancer types, underscoring its broad clinical applicability.
3. **Orthogonal validation of virtual spatial proteomics.** We have validated the HEX model using paired IHC + H&E for predicting protein marker expression. HEX-derived protein maps showed strong concordance with IHC measurements, demonstrating its generalizability across experimental data modalities.
4. **Translational impact on treatment decision-making.** We assessed how HEX-MICA could improve treatment selection in immunotherapy. For example, HEX identified 40% of PD-L1-high patients for whom standard ICI is insufficient and may instead benefit from chemo-ICI combination. Patients with *EGFR*-mutant tumors are considered refractory and typically do not receive ICIs. However, HEX revealed 29% of these patients who may benefit from ICIs.

5. **Advantages over spatial transcriptomics (ST) approaches.** We performed head-to-head comparisons with alternative ST methods and showed that HEX-derived virtual spatial proteomics outperforms state-of-the-art ST models in both gene/marker expression prediction and clinical outcome prediction.

In summary, we have performed extensive new experiments and analyses to demonstrate the generalizability, scalability, and translational impact of our approach. These new results are described below and presented in **17 new Extended Data Figures**: Fig. 4, 5, 6, 7, 8, 9, 11, 23, 25, 30, 31, 32, 35, 36, 37, 38, and 39.

Index to response by reviewers

Response to Reviewer #1.....	Page 3
Response to Reviewer #3.....	Page 17
Response to Reviewer #4.....	Page 30

Reviewer #1 (Remarks to the Author):

The authors set out to create an AI model (HEX) capable of spatially resolved protein expression prediction from paired H&E images by leveraging their previous foundation model for H&E expression, MUSK. They use their own cohort of 10 patients' samples with NSCLC using the PCF v2 system. It is important to note that the current model does not contain the authors' pre-trained model head, which would be needed for validation. They evaluate 40 proteins, and independently validate it on TMA and Tissue Array (TA). A subsequent Transformer NN leverages H&E-based and CODEX embeddings (HEX into DINOv2), is then utilized for finetuning on downstream tasks. Additionally, they utilize Integrated Gradients (IG) to identify tiles of high predictive value to downstream clinical characteristics. They compare different foundation model backbones as well as conduct ablation studies to underscore their architectural choices.

Response: Thank you for your comment about our code availability. We have updated our public repository (<https://github.com/lilab-stanford/HEX>) to include a pre-trained three-layer regression head (`hex_architecture.py`, `checkpoint.pth`). These additions will allow other researchers to verify model input/outputs and fine-tune models for generating virtual spatial proteomics data from H&E.

Major revisions - It is important that further analyses be carried to elucidate some key points:

1. Testing the model on 10 patients only does not necessarily guarantee the robustness of the metrics. Can the authors test expanding or subsetting their dataset has effects on their metrics (PCC, MSE, etc.) in order to establish scaling laws? Similar scaling should be derived for the relationships of the number of markers and downstream clinically relevant results. It is important to also test bench protocol differences, such as the sensitivity of the model to separate H&E staining protocols and scanners, which often differ between institutions.

Response: Thank you for these valuable suggestions. We have conducted additional analyses to thoroughly evaluate the scalability and robustness of our HEX model.

First, to investigate the impact of training dataset size on protein prediction accuracy, we trained HEX using incrementally larger subsets of the Stanford-WSI dataset (2, 4, 8, and 10 WSIs; Extended Data Fig. 37). It is important to note that the model was trained at the image patch-level rather than at WSI level. Since we used matched histopathology and spatial proteomics data, each WSI comprises a large number of patches (75,500 on average) with individually labeled protein expressions, resulting in a substantial training dataset at the patch level.

Our analysis demonstrated that the accuracy of protein prediction increases with more WSIs used for training. We observed significant improvements when expanding from 2 to 4 WSIs (Δ Pearson $r = +0.211$), with diminishing gains from 8 to 10 WSIs (Δ Pearson $r = +0.013$). This indicates that a scaling law does exist within our dataset, where the protein prediction accuracy increases with larger training datasets. With a sufficiently large training set, the improvement in model performance seems to be incremental beyond 500,000 image patches.

Extended Data Fig. 37 | Effect of training dataset size on HEX performance. HEX was trained on 2, 4, 8, and 10 WSIs and evaluated on the Stanford-TMA and TA-TMA cohorts. Performance was assessed using Pearson correlation, Spearman correlation, SSIM, and MSE. Error bars indicate 95% confidence intervals from 1,000 bootstrap replicates.

Next, to assess how the number of protein markers impacts downstream clinical predictions, we constructed reduced marker panels of size $K = 5, 10, \text{ and } 20$, where the markers were selected based on their association with outcome in the NLST training cohort. We then re-trained the MICA prognostic model on NLST and evaluated performance on four independent validation cohorts (TCGA-LUNG, PLCO-LUNG, Stanford-TMA, and TA-TMA), as well as via cross validation on the Stanford immunotherapy cohort.

Across these cohorts, we observed a clear scaling law where the prognostic accuracy increases with larger panel sizes from 5 to 40 markers (Extended Data Fig. 38). Compared with H&E alone, models using the first five protein markers improved the C-index by 7%, the next five by an additional 4% across validation cohorts. Beyond 20 markers, the models showed incremental but diminishing gains in performance.

Extended Data Fig. 38 | Effect of marker panel size on prognostic performance. MICA was trained with 5, 10, 20, and 40 HEX-inferred markers and evaluated across five cohorts. Prognostic accuracy (c-index) improved with panel size but showed diminishing returns beyond 20 markers. Error bars indicate 95% CIs.

Finally, we assessed the robustness of the HEX model to variations in H&E staining protocols and scanners by testing on externally generated datasets. Please see our detailed response to your Comment #2 below.

Revised Manuscript:

Results - Scaling laws (lines 476-500)

2. In the same vein, the authors should validate this on a similarly designed cohort from another tissue, retraining their model to demonstrate generalizability. This need not be newly generated data but can be done using public consortia data too, such as HuBMAP (healthy intestine) and HTAN (BrCa).

Response: Thank you for this important suggestion. To more rigorously evaluate HEX's generalizability to new tissue types and its robustness to variations in histological protocols, we conducted external validation on a publicly available pan-cancer dataset

containing 57-plex CODEX and H&E images of the same section [1]. This dataset comprises 206 TMA cores spanning 34 distinct tissue types, including malignant tumors (e.g., colorectal, breast, kidney, stomach, liver, and pancreas), benign neoplasms, and matched normal tissues. The samples were obtained from the University of Bern in Switzerland and prepared using a different H&E staining protocol and digitized using a Keyence BZ-X710 scanner.

When directly applied to the external Bern dataset, HEX achieved strong predictive accuracy across 24 overlapping protein markers **without retraining or fine-tuning** (mean Pearson $r = 0.658$), which is only slightly lower than in internal validation (mean Pearson $r = 0.718$) (Extended Data Fig. 5). Further, we show that HEX consistently and substantially outperformed the second-best model, CGAN, across all key metrics: Pearson r (0.658 vs. 0.210), Spearman r (0.563 vs. 0.140), SSIM (0.638 vs. 0.521), and MSE (0.132 vs. 0.835). These results confirm HEX's generalizability to new tissue types and robustness to variability in staining protocols.

Extended Data Fig. 5 | External validation of HEX on the Bern pan-cancer dataset. a, Pearson correlation between predicted and measured protein expression across 24 overlapping markers. **b,** Comparison of HEX performance between two imaging platforms: Leica AT2 (Stanford-TMA and TA-TMA) and Keyence BZ-X710 (Bern). **c,** Comparison of HEX with baseline models on the Bern dataset. Error bars indicate 95% CIs from 1,000 bootstrap replicates.

To further test generalizability through retraining, we focused on a specific tissue subset—140 colorectal cancer (CRC) cores in the Bern dataset—and trained HEX de novo using all 57 available protein markers, including 33 that were not present in the

original NSCLC panel. We used 84 cores from 21 patients for training and 56 cores from 14 patients for testing. The retrained model achieved a good performance with mean Pearson r = 0.566 across 57 markers (Extended Data Fig. 6), confirming HEX's capacity to generalize when trained from scratch on a different tissue type.

Extended Data Fig. 6 | HEX retrained on colorectal cancer. Pearson correlation for 57 markers after retraining HEX on CRC. Orange bars denote markers overlapping with the NSCLC panel; blue bars are CRC-specific. Error bars show 95% CIs.

Revised Manuscript:

Results - HEX enables accurate prediction of protein expression from H&E images - Independent validation performance (lines 169-190)

3. Given that this model is trained on NSCLC, is the finetuning of this model possible to other tissues, given a similar cohort?

Response: Yes. We fine-tuned the NSCLC-trained HEX model on the same 140 CRC cores used in our retraining experiment, using 84 cores from 21 patients for training and 56 cores from 14 patients for testing. Fine-tuning was performed by reinitializing the output head and adapting the model to the CRC data. This yielded improved predictive performance (mean Pearson r = 0.659 vs. 0.566 with retraining from scratch; Extended Data Fig. 7). These results demonstrate that HEX can be efficiently transferred to new tissues with limited data and minimal architectural changes, offering a practical strategy for adapting virtual spatial proteomics to diverse tumor types.

Extended Data Fig. 7 | HEX fine-tuned on colorectal cancer. Pearson correlation for 57 markers after fine-tuning the NSCLC-trained HEX model on CRC. Orange bars indicate markers overlapping with the NSCLC panel; blue bars are CRC-specific. Error bars show 95% CIs.

Revised Manuscript:

Results - HEX enables accurate prediction of protein expression from H&E images - Independent validation performance (lines 191-192)

4. Given that this model is trained on 40 specific markers, is this model amenable to a different set of markers, supposing the head is changed?

Response: Yes. As described in our responses to Comments 2 and 3 above, we evaluated HEX’s ability to generalize to a 57-marker CRC panel that includes 33 novel proteins not present in the original NSCLC panel. By reinitializing the output head, HEX readily adapted to this expanded panel, where the performance was even slightly better for predicting novel protein markers than in the legacy panel. Fine-tuning the NSCLC-trained model yielded stronger performance than retraining from scratch (mean Pearson $r = 0.659$ vs. 0.566). As shown in Extended Data Fig. 8, these improvements were consistent across both legacy (overlapping) and novel (CRC-specific) markers, confirming that HEX can be efficiently extended to alternative marker panels through transfer learning.

Extended Data Fig. 8 | Fine-tuning improves prediction across legacy and novel markers. Performance comparison on legacy (24) and novel (33) markers after retraining or fine-tuning. Error bars show 95% CIs.

Revised Manuscript:

Results - HEX enables accurate prediction of protein expression from H&E images - Independent validation performance (lines 193-197)

5. How much better is the predictive accuracy of HEX to just using MUSK (or another foundation model) – e.g. in the MUSK image + report or MUSK image cases from the authors’ previous publication?

Response: Thank you for this important question. HEX builds on the MUSK foundation model by introducing a multi-modal learning objective for spatial proteomics and specialized training strategies tailored to protein inference from histology.

As detailed in the manuscript, we have already compared HEX against MUSK in both protein-level prediction and downstream clinical tasks. For protein prediction, we conducted ablation experiments by removing HEX-specific training strategies, effectively reducing the model to a MUSK-only baseline (Extended Data Fig. 12–13). For prognosis and immunotherapy response prediction in NSCLC, we implemented the H&E-only baseline using MUSK features with AbMIL—equivalent to the “MUSK image” model in our previous work. All these comparisons demonstrate superior performance of HEX compared with standard-alone MUSK, which are presented in Fig. 4b, Fig. 6a, and Fig. 6b.

To assess broader clinical relevance, we extended prognostic modeling to 12 additional cancer types in TCGA (n = 5,019 patients). We compared three different models: (i) a MUSK-based H&E-only model, (ii) a virtual CODEX-only model, and (iii) the full MICA model integrating both. MICA consistently outperformed both unimodal baseline models across all cancer types (mean c-index: 0.73 vs. 0.67 for H&E-only model and 0.66 for virtual CODEX-only model; P < 0.001; Extended Data Fig. 25). These new results

confirm that HEX can significantly improve upon the original MUSK model in pan-cancer prognosis prediction.

Extended Data Fig. 25 | MICA improves pan-cancer prognosis prediction. a, C-index comparison of H&E-only (MUSK), virtual CODEX-only, and MICA models. MICA outperforms both baselines across all cancer types. Error bars show standard deviation. **b**, Kaplan-Meier curves for 12 TCGA cancer types using MICA-predicted risk groups.

Revised Manuscript:

Results - HEX improves pan-cancer prognosis prediction (lines 289-305)

Methods - Training and evaluation of MICA (lines 892-898, 907-910)

6. Similarly, there should be a head to head comparison of c-indices that includes the real CODEX data, where it is available (Stanford-TMA and TA-TMA). In every head to head comparison, proper statistical testing should be performed to assess if a difference in performance between any two methods is statistically significant.

Response: Thank you for this valuable suggestion. We have conducted a head-to-head comparison of prognostic performance using true CODEX data (MICA-True), virtual CODEX data generated by HEX (MICA-Virtual), and H&E-only features in the two cohorts with matched true spatial proteomics: Stanford-TMA and TA-TMA.

All models were trained in the NLST cohort and evaluated on the two TMA cohorts. As shown in Extended Data Fig. 23, both MICA-True and MICA-Virtual models significantly outperformed the H&E-only model. Importantly, there were no statistically significant difference in C-index between MICA-True and MICA-Virtual in either cohort. These results demonstrate that HEX-derived virtual CODEX can provide additional prognostic information beyond H&E that's equivalent to true CODEX data. Our findings support the use of HEX as a practical and scalable alternative to experimental spatial proteomic profiling in the context of prognosis prediction.

Extended Data Fig. 23 | HEX-derived virtual spatial proteomics matches true spatial proteomics for prognosis prediction. Comparison of C-index for H&E-only, MICA-Virtual (HEX), and MICA-True (true CODEX) models on Stanford-TMA and TA-TMA cohorts. MICA-Virtual significantly outperforms H&E-only, and performs comparably to MICA-True (n.s.). Error bars indicate 95% CIs.

Revised Manuscript:

Results - HEX improves prognosis prediction in early-stage lung cancer (lines 271-280)

7. The authors should include H-Optimus-v1 (or v0) to the comparisons for backbones, as it is the most popular model on HuggingFace and widely used by the community.

Response: We have extended our analysis of foundation model backbones to include H-Optimus-v1, one of the most widely used models in the community. As shown in Extended Data Fig. 11, HEX models initialized from the H-Optimus-v1 backbone achieved competitive performance for protein expression prediction from H&E. These results confirm the general applicability of the HEX framework across multiple pathology foundation models, while supporting our choice of MUSK as the most performant backbone in this application.

Extended Data Fig. 11 | Performance comparison of foundation model backbones for HEX on independent validation cohorts. MUSK-based HEX consistently outperformed other backbones, achieving the highest predictive accuracy across external datasets. Models based on CONCH, UNI, Virchow2, and H-optimus-1 showed varying levels of performance and speed, reflecting different trade-offs between accuracy and computational efficiency.

8. In the SCS analysis, how were those specific two marker signatures chosen? If all of them were tested, are the provided p-values for those specific ones adjusted for multiple hypothesis testing?

Response: For the first question, regarding the selection of marker signatures in the SCS analysis, the two highlighted marker pairs were chosen from a **pre-specified** panel of six biologically informed combinations. Each pair consists of a canonical cell lineage marker and a functional/phenotypical marker, allowing us to interrogate both cell identity

and cell state in the spatial context. The six pairs and their associated cell types are as follows:

1. Granzyme B⁺ CD8⁺: activated cytotoxic T cells
2. TCF-1⁺ CD4⁺: naïve or stem-like CD4⁺ T cells
3. MMP9⁺ CD66b⁺: tumor-associated neutrophils
4. PD-1⁺ CD8⁺: dysfunctional or progenitor-like CD8⁺ T cells
5. Collagen IV⁺ FAP⁺: matrix-producing cancer-associated fibroblasts (mCAFs)
6. MMP9⁺ CD163⁺: matrix-remodeling tumor-associated macrophages (TAMs)

These pairs were selected based on their established relevance to immunotherapy response, such as Granzyme B⁺ CD8⁺ T cells [2] and PD-1⁺ CD8⁺ T cells [3], as well as their potential roles in immunosuppressive regulation within the tumor microenvironment, including MMP9⁺ neutrophils [4] and MMP9⁺ tumor-associated macrophages [5]. Additional phenotypes of interest included TCF-1⁺ CD4⁺ T cells [6] and mCAFs [7]. Our aim was to assess whether the spatial proximity between these phenotypically distinct cell populations, measured using the Jaccard index, is associated with immunotherapy outcomes in the ICB cohort.

For the second question, regarding multiple hypothesis testing, this analysis was restricted to a small number of pre-specified and biologically informed hypotheses (n = 6). Therefore, we report the nominal *P*-values without correction. This rationale is now explicitly described in both the Results and Methods sections.

Revised Manuscript:

Results - Biological interpretation of HEX-based immunotherapy response prediction (lines 423-425)

Methods - Biological interpretation of outcome predictions (lines 936-941)

Minor Revisions - Additionally, there are some minor concerns that the authors should clarify:

1. PCF scanned at 40x magnification, but MUSK has been trained at 20x magnification (as have most foundation model), is this something that is addressed through PALOM registration?

Response: While many foundation models are trained at 20x magnification, the MUSK backbone used in HEX was pretrained using multi-scale image augmentations spanning

10x, 20x, and 40x fields of view, making it robust to variations in image scale. In our pipeline, all H&E whole-slide images were processed at 40x resolution (~0.25 $\mu\text{m}/\text{pixel}$). Slides scanned at lower magnifications were rescaled to 40x to maintain consistency. PALOM registration was only applied to align H&E and CODEX images, both at 40x resolution, and does not play a role in resolving magnification mismatches. We have clarified this point in the revised Methods section (“WSI processing”, lines 741-743).

2. Are there any trends that relate to protein localization? Nuclear, cytoplasmic, membrane and correlations?

Response: Thanks for this insightful question. We divided the 40 protein markers by subcellular localization—nuclear, cytoplasmic, and membrane—and calculated average Pearson correlations for each category in both cross-validation and independent validation settings. As shown in Extended Data Fig. 4, HEX performed similarly across all three compartments, with slightly higher accuracy for nuclear markers. These results indicate that HEX generalizes well across proteins with diverse localization patterns. This analysis has been added to the revised manuscript.

Extended Data Fig. 4 | HEX protein prediction accuracy by subcellular localization. Average Pearson correlations for nuclear, cytoplasmic, and membrane markers in cross-validation (left) and independent validation (right). Error bars indicate standard deviation (left) and 95% CIs (right).

Revised Manuscript:

Results - HEX enables accurate prediction of protein expression from H&E images - Independent validation performance (lines 162-167)

3. The published version of MUSK uses 384x384 patches, but here 224x224 are listed. Was the model reconfigured for this task or just the tiles resized prior to going through the model?

Response: In our pipeline, image patches were initially extracted at 224×224 pixels and then resized to 384×384 before being passed into the MUSK backbone, matching the expected input resolution. No reconfiguration of the MUSK architecture is necessary. As previously noted, MUSK is pretrained with multi-scale augmentation and applicable to images with various magnifications.

4. Were the H&Es normalized in any way? Similarly for the CODEX data.

Response: For H&E images, we applied standard pixel-wise normalization (mean = 0.5, std = 0.5) to match the MUSK backbone’s expected input distribution. For CODEX data, we normalized each marker channel by scaling intensities to the [0, 1] range based on the 99th percentile of the per-marker distribution. These procedures have been clarified in the revised Methods section (“WSI processing” , lines 745-749).

5. The Mann-Whitney U test in Extended Data Fig. 22 might be skewed by the large number of patches. It would be great to also note effect sizes for markers there or test against a specific effect size or report similar metrics like Cohen’s d.

Response: Thank you for this suggestion. In the revised Extended Data Fig. 32, we have included effect sizes using the rank-biserial correlation, which is appropriate for Mann–Whitney U tests. This provides additional information about group differences.

Extended Data Fig. 32 | Box plots comparing marker expression distributions between the high- and low-risk groups for all protein biomarkers. Statistical significance was assessed using the two-sided Mann–Whitney U test. Effect sizes were calculated using rank-biserial correlation, providing a standardized measure of group differences. ***, $P < 0.001$.

6. Regarding comparisons with VirtualMultiplexer and DeepLIIF, were they pretrained to predict the same 40 markers? Or used only the intersecting ones with this dataset?

Response: For fair and consistent comparison, both VirtualMultiplexer and DeepLIIF were pretrained to predict the same 40 protein markers as HEX on our matched CODEX-

H&E dataset using their respective publicly available implementations and training protocols. This has been clarified in the revised Methods - HEX model evaluation and comparison (lines 841-844).

7. Some notes should be made on inference time and compute resource requirements. Some of these are available in the GitHub but should also be noted in the Supplementary.

Response: Thanks for the suggestion. We have added a summary of inference time and compute requirements in the revised Methods - HEX model training details (lines 817-819). On standard hardware (8 × NVIDIA L40S GPUs), inference takes ~1.3 minutes per WSI at 40x magnification with a batch size of 16. HEX also supports single-GPU inference with ≥ 8 GB VRAM. This information has been included in the Methods to improve clarity on the model's computational efficiency.

8. Which DINO model was used? (It seems it is dinov2_vits14_reg but this should be mentioned in the text).

Response: Yes, we used the dinov2_vits14_reg model for feature extraction. This has now been explicitly stated in the revised Methods - Training and evaluation of MICA (lines 880-882).

Reference (Reviewer #1):

1. Schürch, C.M., et al. Coordinated Cellular Neighborhoods Orchestrate Antitumoral Immunity at the Colorectal Cancer Invasive Front. *Cell* 182, 1341-1359.e1319 (2020).
2. Giles, J.R., Globig, A.-M., Kaech, S.M. & Wherry, E.J. CD8+ T cells in the cancer-immunity cycle. *Immunity* 56, 2231-2253 (2023).
3. Damo, M., et al. PD-1 maintains CD8 T cell tolerance towards cutaneous neoantigens. *Nature* 619, 151-159 (2023).
4. Zilionis, R., et al. Single-cell transcriptomics of human and mouse lung cancers reveals conserved myeloid populations across individuals and species. *Immunity* 50, 1317-1334.e1310 (2019).
5. Lu, Y., et al. A single-cell atlas of the multicellular ecosystem of primary and metastatic hepatocellular carcinoma. *Nature communications* 13, 4594 (2022).
6. Zou, D., et al. CD4+ T cell immunity is dependent on an intrinsic stem-like program. *Nature immunology* 25, 66-76 (2024).
7. Zhang, H., et al. Define cancer-associated fibroblasts (CAFs) in the tumor microenvironment: new opportunities in cancer immunotherapy and advances in clinical trials. *Molecular cancer* 22, 159 (2023).

Reviewer #3 (Remarks to the Author):

Summary of the key results

Spatial proteomics is a relatively novel technology that allows the detection of multiple proteins on a tissue section. Authors have developed an AI model designated as HEX (H&E to protein eXpression), that is able to generate spatial proteomics profiles from standard H&E slides. They selected a total of 40 targets that include immune, stromal and tumor cells. A multimodal data integration approach that combines the H&E and the AI-derived spatial proteomics was shown to enhance outcome prediction. Authors conclude that HEX is a low-cost approach that provides interpretable biomarkers for precision medicine.

Originality and significance: if not novel, please include reference

Other studies have developed AI models able to predict cell types or proteins from HE examples: PMID: 40097393, PMID: 32747659

Response: Thank you for pointing out to these two relevant studies which we have now included in our citations. Schmauch et al. (PMID 32747659) tried to predict **bulk gene expression** from histology images, which suffers from both technical and fundamental challenges due to intra-tumor heterogeneity. Jasti et al. (PMID 40097393) aims to predict spatial protein expression from H&E; however, the prediction is only done for a **single protein marker (CD31)**, which limits the biological scope and clinical utility.

Similar literatures have already been discussed in the introduction and their limitations are clearly described as compared to our approach, which is aimed at predicting high-plex CODEX-based spatial proteomics from H&E using pathology foundation models.

Compared with prior studies, our HEX framework introduces four key advances:

1. **High-quality training data with matched spatial proteomics and histology:** HEX is trained on paired 40-plex CODEX and H&E acquired on the **same section**, and using state-of-the-art pathology foundation models enables accurate prediction of spatially resolved protein expression (Pearson $r = 0.73-0.79$). This is a significant advance over existing studies trying to predict a single protein marker (Jasti et al.) or RNA expression at the bulk level (Schmauch et al.) with a much lower accuracy (Pearson $r = 0.25$).
2. **High-plex prediction enabling biological insight:** HEX simultaneously infers a diverse panel of 40 protein markers spanning immune, stromal, and tumor programs. This broad coverage is critical, as it allowed us to uncover novel, clinically relevant spatial co-localization signatures that are impossible to interrogate with bulk RNA signature or single-marker models.

3. **Multimodal integration for superior outcome prediction:** We designed a novel co-attention fusion network that effectively combines virtual spatial proteomics with H&E images, which consistently outperforms either modality alone for prognosis and immunotherapy-response prediction. This type of multimodal integration has not been explored in prior studies.
4. **Extensive clinical validation in pan-cancer setting:** We evaluated our HEX framework in six independent cohorts of over 2,000 patients with NSCLC as well as over 5,000 patients across 12 additional cancer types (see new data below). Importantly, we show that HEX-derived virtual spatial proteomics matches the performance of real CODEX for prognosis prediction (Extended Data Fig. 23).

Overall, these represent a substantive methodological and translational advance over existing studies including the two mentioned here. We have clarified this in the revision.

Extended Data Fig. 23 | HEX-derived virtual spatial proteomics matches true spatial proteomics for prognosis prediction. Comparison of C-index for H&E-only, MICA-Virtual (HEX), and MICA-True (true CODEX) models on Stanford-TMA and TA-TMA cohorts. MICA-Virtual significantly outperforms H&E-only, and performs comparably to MICA-True (n.s.). Error bars indicate 95% CIs.

Data & methodology: validity of approach, quality of data, quality of presentation
The paper is well written and straightforward

Appropriate use of statistics and treatment of uncertainties
Methods appear solid.

Conclusions: robustness, validity, reliability
-The biological interpretation of the model remain superficial, as the study is purely descriptive. To be further explored functional experiments should be further conducted.

Response: We appreciate your comment here. First, we wish to clarify that the primary purpose of the study is to **develop an enabling technology** for AI-enabled virtual spatial proteomics from standard histopathology, which can then be used for spatial biomarker discovery on a large scale. Indeed, we have shown through extensive independent validation that histology-based virtual spatial proteomics is feasible and effective for predicting clinical outcomes in more than 7,000 patients across 13 cancer types. The purpose and design of technology-development studies like ours is different from traditional hypothesis-driven studies focused on experimental validation of specific biological findings.

In this study, to assess the biological interpretation and relevance of HEX-derived predictions of immune checkpoint therapy response, we focused on cellular subsets which are defined by a lineage-defining marker with a functional or state-associated marker. While not experimentally validated in this study, below we briefly outline the mechanistic evidence and biological relevance of HEX-nominated candidate cell types which are supported by functional experiments conducted in prior studies.

1. Granzyme B⁺ CD8⁺ T cells: These represent activated cytotoxic T lymphocytes (CTLs), Granzyme B is a key effector molecule delivered via perforin-mediated pores to induce apoptosis in antigen-expressing tumor cells. Tumor cell killing, cytolytic granule release, and its correlation with immune checkpoint therapy response have been well documented through functional experiments [1, 2, 3].

2. TCF-1⁺ CD4⁺ T cells: This population represents naïve or stem-like helper T cells that exhibit memory-like or progenitor characteristics and are capable of sustaining long-term immunity under chronic antigenic stimulation. These cells have been shown to preserve proliferative capacity and contribute to the maintenance of functional CD8⁺ T cell responses in both autoimmunity and cancer. Their relevance to immunotherapy efficacy has been demonstrated in preclinical tumor models and clinical cohorts [4]. Furthermore, the Tcf7 locus, which encodes TCF-1, has been shown to undergo de novo epigenetic reprogramming in circulating autoimmune CD4⁺ T cells, supporting its role as a developmentally pre-programmed regulator of long-lived CD4⁺ T cell states [5]. These findings highlight the functional importance of TCF-1⁺ CD4⁺ T cells in both immune memory and therapeutic response settings.

3. MMP9⁺ CD66b⁺ neutrophils: This cellular subset corresponds to tumor-associated neutrophils (TANs) characterized by matrix remodeling and pro-tumorigenic properties. Functionally, MMP9 contributes to tumor progression by promoting angiogenesis, facilitating immune evasion, and supporting metastatic dissemination [6, 7, 8, 9, 10]. Recent studies have further identified MMP9⁺ CXCR2⁺ TANs as a distinct immunosuppressive state enriched in the tumor microenvironment of both human tumors and mouse models [11].

4. PD-1⁺ CD8⁺ T cells: These include exhausted or progenitor-like CD8⁺ T cells. Their ability to respond to checkpoint blockade is well documented, and their presence in tumors predicts therapeutic response depending on their differentiation state and TCF1 co-expression [12, 13].

5. FAP⁺ Collagen IV⁺ CAFs: This population represents matrix-producing cancer-associated fibroblasts (mCAFs) that play a central role in stromal remodeling and immune exclusion. These cells contribute to therapy resistance by constructing both physical extracellular matrix (ECM) barriers and establishing immunosuppressive chemical gradients that impede T cell infiltration [14, 15, 16]. Functional studies have demonstrated that FAP⁺ CAFs deposit dense collagen networks and modulate the tumor microenvironment to favor immune evasion, thereby limiting the efficacy of immunotherapy [17].

6. MMP9⁺ CD163⁺ macrophages: This subset of tumor-associated macrophages (TAMs) exhibits potent matrix-remodeling and immunosuppressive activity. Through MMP9 secretion, a key member of the matrix metalloproteinase family, these cells promote extracellular matrix degradation and contribute to the formation of a suppressive tumor microenvironment that impedes effector T cell infiltration [18, 19]. Recent studies in CTNNB1-mutant hepatocellular carcinoma further demonstrate that pharmacological inhibition of MMP9 reactivates CD8⁺ T cell-mediated antitumor immunity and enhances the efficacy of anti-PD-1 therapy [20]. These findings underscore the mechanistic importance of MMP9⁺ TAMs in driving immune evasion and immunotherapy resistance.

Beyond the single-cell phenotypes described above, we computed a spatial proximity (Jaccard index) to quantify the pairwise cell-cell co-localization patterns and evaluate its association with immunotherapy response. For the cell-cell correlation matrix shown in Fig. 6e, several lines of evidence, including functional studies, support the biological plausibility of our findings. Specifically, the spatial proximity between PD1⁺ CD8⁺ T cells and Granzyme B⁺ CD8⁺ T cells was significantly elevated in the responders vs non-responders, suggesting that co-localization of exhausted or progenitor-like CD8⁺ T cells with cytotoxic effector CD8⁺ T cells may reflect an ongoing phenotypic transition within the tumor microenvironment. This spatial organization could enable effective reinvigoration of T cell cytotoxicity upon immune checkpoint blockade. Prior studies have shown that PD1 blockade promotes the differentiation of PD1⁺ progenitor CD8⁺ T cells into functional cytotoxic lymphocytes, thereby improving therapeutic efficacy [21, 22].

Overall, these findings support that the cell phenotypes implicated by our HEX framework are independently validated through in vivo and in vitro functional experiments. Moreover, the spatial cell-cell interactions uncovered by HEX are

experimentally supported and mechanistically linked to immunotherapy response and clinical outcomes. We have discussed this in the revised Discussion section (lines 564-577).

Suggested improvements: experiments, data for possible revision

-The sentence in the abstract "trained on 755,000 histopathology images with matched protein expression" as the number of different samples is much less than 755000

Response: Thank you for this comment. We have revised the sentence to clarify the number of individual image tiles as well as the number of samples used in our study. The revised sentence now reads:

'Trained and validated on 819,000 histopathology image tiles with matched protein expression from 382 tumor samples, HEX accurately predicts the expression of 40 biomarkers encompassing immune, structural, and functional programs. HEX is further extended to 34 tissue types and demonstrates substantial performance gains over alternative methods for protein expression prediction from H&E images.'

-The first part of the results (first paragraph-study overview) is not really needed

Response: We have removed the initial study overview paragraph from the Results section to streamline the presentation and avoid redundancy with the Introduction and Methods.

-A significant proportion of the c indexes obtained are below 0.8, which limits the use of such models in practice

Response: We agree with you that a c-index higher than 0.8 for prognosis prediction will lead to a larger clinical impact. However, this is not an absolute requirement for demonstrating clinical utility. The key question is **how much improvement can be achieved compared with current standard of care**. For example, TNM staging is routinely used to inform adjuvant therapy after curative-intent surgery. In our study, the average c-index for staging is only 0.58 across the four lung cancer cohorts. By contrast, integrating HEX-derived risk scores with clinical factors significantly enhanced prognostic accuracy (overall c-index 0.71), which is an increase of 22% compared with staging. Although not perfect, this is expected to improve risk stratification and may reduce under- and over-treatment with adjuvant therapy that is currently informed by staging.

Similarly, in the context of immunotherapy response, standard clinical biomarker PD-L1 had a modest predictive value with AUC of 0.66. Tumor mutation burden (TMB), another FDA-approved test, had an even lower AUC of 0.59. By comparison, our HEX approach achieved significantly higher accuracy with an AUC of 0.82, which represents a 24-39% improvement.

To better contextualize how the HEX–MICA model can influence clinical decision-making, we conducted additional analyses beyond conventional performance metrics.

First, we assessed its ability to improve patient stratification for immunotherapy response. Compared to standard biomarker PD-L1, the HEX model correctly reclassified 4 out of 35 (11%) patients as true responders; at the same time, HEX also correctly reclassified 6 out of 55 (11%) patients as true non-responders, resulting in a positive Net Reclassification Index of 22%.

Second, we compared the HEX model to standard biomarkers including PD-L1 and TMB in terms of sensitivity at a fixed specificity of 90% for predicting objective response. Our model achieved a sensitivity of 61%, substantially outperforming PD-L1 (19%) and TMB (10%) (Extended Data Fig. 30). This analysis underscores the potential of HEX model to better distinguish true responders from non-responders, thereby improving the effective response rates while sparing patients of unnecessary treatment.

We then assessed how the new model would improve treatment decisions in the context of current clinical practice. ICI monotherapy is a standard of care in NSCLC patients with tumor PD-L1 $\geq 50\%$ given the likelihood of strong response. However, our HEX model identified 19 out of 47 (40%) patients with PD-L1-high tumors who did not experience durable response to ICI, with a median PFS interval of 4 months vs. 20 months in the predicted responder group (Fig. 6c). These patients may benefit from concurrent chemotherapy in addition to ICI. For patients with intermediate levels of PD-L1 ($>1\%$ and $<50\%$), the HEX model significantly stratified patients for PFS (HR = 3.57, P = 0.0009) and thus could also help optimize the treatment decisions on ICI monotherapy vs. combination ICI and chemotherapy.

On the other hand, for patients with tumor PD-L1 = 0%, there is a moderate benefit of anti-CTLA/anti-PD1 ICIs and chemotherapy over standard anti-PD1 ICIs and chemotherapy [23]. However, the triplet combination therapy has increased toxicity, and the optimal regimen is unclear at the patient level. The HEX model identified 23 out of 38 (61%) patients with PD-L1-negative tumors who did not benefit from anti-PD1 ICIs, with a median PFS interval of 4 months vs. 13 months in the predicted responder group (Fig. 6c). These patients may benefit from novel combination treatment strategies.

Finally, patients with EGFR-mutant tumors are generally considered refractory to ICIs and therefore do not receive ICIs as their standard treatment. In our analysis of the

immunotherapy cohort, the HEX mode identified 5 out of 17 (29%) of patients whose tumors harbor EGFR mutation but derived durable clinical benefit from ICIs, with a median PFS interval of 8 months vs. 4 months in the predicted non-responder group (Extended Data Fig. 31).

Taken together, these results suggest that virtual spatial proteomics and multimodal modeling with HEX–MICA can provide meaningful improvements in patient stratification, with the potential to enhance treatment selection and improve patient outcomes.

Extended Data Fig. 30 | Sensitivity at fixed specificity for predicting immunotherapy response. Comparison of MICA, PD-L1, and TMB for predicting objective response at 90% specificity.

Extended Data Fig. 31 | Kaplan–Meier analysis of MICA risk groups in Stanford-IO subgroups defined by EGFR. P values were calculated using two-sided log-rank tests.

Revised Manuscript:

Results - HEX improves immunotherapy response prediction in advanced lung cancer (lines 394-408)

Discussion (lines 541-557)

-Given that the vast majority of the markers used are not lung-specific, investigation in other cohorts of patients with cancers from other organs would be interesting

Response: Thank you for raising this important question. To evaluate the broader applicability of HEX beyond lung cancer, we have conducted a series of experiments across multiple datasets, cancer types, and different biomarker panels to assess the generalizability of HEX.

1. **External validation across diverse tissues**

We directly applied the NSCLC-trained HEX model—without any retraining or fine-tuning—to a pan-cancer dataset with paired 57-plex CODEX and H&E images from the University of Bern. This publicly available dataset includes 206 TMA cores spanning 34 tissue types, including malignant tumors (e.g., breast, colorectal, stomach, liver, pancreas, kidney, bladder, etc.), benign neoplasms, and normal tissues, prepared using different staining protocols and scanned on a different imaging platform [24]. Despite these technical variations, HEX achieved strong protein prediction performance across 24 overlapping biomarkers (mean Pearson $r = 0.658$, Extended Data Fig. 5), only slightly lower than on the original Stanford dataset ($r = 0.718$). Further, HEX substantially outperformed the second-best model, CGAN, consistently across all key metrics: Pearson r (0.658 vs. 0.210), Spearman r (0.563 vs. 0.140), SSIM (0.638 vs. 0.521), and MSE (0.132 vs. 0.835), confirming its generalizability across tissue types, staining protocols, and imaging platforms.

2. **Extension to new protein markers in a different cancer**

To further test adaptability of HEX to new marker panels in other tissues, we performed additional experiments by retraining or finetuning the HEX model to predict all 57 protein markers—including 33 not present in the original NSCLC panel. For this purpose, we used 140 colorectal cancer (CRC) cores from the Bern dataset. When trained de novo, HEX achieved a Pearson correlation of $r = 0.566$ (Extended Data Fig. 6). Fine-tuning the NSCLC-trained model by reinitializing the output head and updating all parameters improved performance to Pearson correlation $r = 0.659$ (Extended Data Fig. 7). These improvements were consistent across both legacy and CRC-specific markers (Extended Data Fig. 8), demonstrating that HEX can be efficiently adapted to other tissue types and expanded marker panels with limited additional training data.

3. **Pan-cancer prognosis prediction using virtual spatial proteomics**

To evaluate clinical utility across cancer types, we extended prognosis prediction to 12 additional cancer types involving 5,019 patients in TCGA. Using virtual proteomic maps generated by HEX, we trained the multimodal MICA model and compared its prognostic performance to two unimodal baselines (H&E-only and

HEX-derived virtual CODEX). MICA consistently outperformed both baselines (mean c-index = 0.732, Extended Data Fig. 25), with significant stratification across all cancer types (log-rank $P \leq 0.0001$). These results confirm that HEX-derived features contribute to clinically meaningful predictions across histologically diverse malignancies.

Taken together, these additional analyses demonstrate that HEX is broadly generalizable across tissue types, cancer types, and biomarker panels, and can be efficiently adapted to new clinical settings with limited training data.

Extended Data Fig. 5 | External validation of HEX on the Bern pan-cancer dataset. a, Pearson correlation between predicted and measured protein expression across 24 overlapping markers. **b**, Comparison of HEX performance between two imaging platforms: Leica AT2 (Stanford-TMA and TA-TMA) and Keyence BZ-X710 (Bern). **c**, Comparison of HEX with baseline models on the Bern dataset. Error bars indicate 95% CIs from 1,000 bootstrap replicates.

Extended Data Fig. 6 | HEX retrained on colorectal cancer. Pearson correlation for 57 markers after retraining HEX on CRC. Orange bars denote markers overlapping with the NSCLC panel; blue bars are CRC-specific. Error bars show 95% CIs.

Extended Data Fig. 7 | HEX fine-tuned on colorectal cancer. Pearson correlation for 57 markers after fine-tuning the NSCLC-trained HEX model on CRC. Orange bars indicate markers overlapping with the NSCLC panel; blue bars are CRC-specific. Error bars show 95% CIs.

Extended Data Fig. 8 | Fine-tuning improves prediction across legacy and novel markers. Performance comparison on legacy (24) and novel (33) markers after retraining or fine-tuning. Error bars show 95% CIs.

Extended Data Fig. 25 | MICA improves pan-cancer prognosis prediction. a, C-index comparison of H&E-only (MUSK), virtual CODEX-only, and MICA models. MICA outperforms both baselines across all cancer types. Error bars show standard deviation. **b**, Kaplan–Meier curves for 12 TCGA cancer types using MICA-predicted risk groups.

Revised Manuscript:

Results - HEX enables accurate prediction of protein expression from H&E images - Independent validation performance (lines 169-197)

Results - HEX improves pan-cancer prognosis prediction (lines 289-305)

Methods - Training and evaluation of MICA (lines 892-898, 907-910)

Reference (Reviewer #3):

1. St Paul, Michael, and Pamela S Ohashi. "The Roles of CD8+ T Cell Subsets in Antitumor Immunity." *Trends in cell biology* vol. 30,9 (2020): 695-704. doi:10.1016/j.tcb.2020.06.003
2. Halle, Stephan et al. "In Vivo Killing Capacity of Cytotoxic T Cells Is Limited and Involves Dynamic Interactions and T Cell Cooperativity." *Immunity* vol. 44,2 (2016): 233-45. doi:10.1016/j.immuni.2016.01.010
3. Weigel, Bettina et al. "Cytotoxic T cells are able to efficiently eliminate cancer cells by additive cytotoxicity." *Nature communications* vol. 12,1 5217. 1 Sep. 2021, doi:10.1038/s41467-021-25282-3
4. Zou, Dawei et al. "CD4+ T cell immunity is dependent on an intrinsic stem-like program." *Nature immunology* vol. 25,1 (2024): 66-76. doi:10.1038/s41590-023-01682-z
5. Aljobaily, Nouf et al. "Autoimmune CD4+ T cells fine-tune TCF1 expression to maintain function and survive persistent antigen exposure during diabetes." *Immunity* vol. 57,11 (2024): 2583-2596.e6. doi:10.1016/j.immuni.2024.09.016
6. Ardi, Veronica C et al. "Human neutrophils uniquely release TIMP-free MMP-9 to provide a potent catalytic stimulator of angiogenesis." *Proceedings of the National Academy of Sciences of the United States of America* vol. 104,51 (2007): 20262-7. doi:10.1073/pnas.0706438104
7. Vannitamby, Amanda et al. "Tumour-associated neutrophils and loss of epithelial PTEN can promote corticosteroid-insensitive MMP-9 expression in the chronically inflamed lung microenvironment." *Thorax* vol. 72,12 (2017): 1140-1143. doi:10.1136/thoraxjnl-2016-209389
8. Coussens, L M et al. "MMP-9 supplied by bone marrow-derived cells contributes to skin carcinogenesis." *Cell* vol. 103,3 (2000): 481-90. doi:10.1016/s0092-8674(00)00139-2
9. Zilionis, Rapolas et al. "Single-Cell Transcriptomics of Human and Mouse Lung Cancers Reveals Conserved Myeloid Populations across Individuals and Species." *Immunity* vol. 50,5 (2019): 1317-1334.e10. doi:10.1016/j.immuni.2019.03.009
10. Xue, Ruidong et al. "Liver tumour immune microenvironment subtypes and neutrophil heterogeneity." *Nature* vol. 612,7938 (2022): 141-147. doi:10.1038/s41586-022-05400-x
11. Wu, Yingcheng et al. "Neutrophil profiling illuminates anti-tumor antigen-presenting potency." *Cell* vol. 187,6 (2024): 1422-1439.e24. doi:10.1016/j.cell.2024.02.005
12. Humblin, Etienne et al. "The costimulatory molecule ICOS limits memory-like properties and function of exhausted PD-1+CD8+ T cells." *Immunity*, S1074-7613(25)00248-1. 1 Jul. 2025, doi:10.1016/j.immuni.2025.06.001
13. Damo, Martina et al. "PD-1 maintains CD8 T cell tolerance towards cutaneous neoantigens." *Nature* vol. 619,7968 (2023): 151-159. doi:10.1038/s41586-023-06217-y

14. Cords, Lena et al. "Cancer-associated fibroblast phenotypes are associated with patient outcome in non-small cell lung cancer." *Cancer cell* vol. 42,3 (2024): 396-412.e5. doi:10.1016/j.ccell.2023.12.021
15. Forsthuber, Agnes et al. "Cancer-associated fibroblast subtypes modulate the tumor-immune microenvironment and are associated with skin cancer malignancy." *Nature communications* vol. 15,1 9678. 8 Nov. 2024, doi:10.1038/s41467-024-53908-9
16. Arpinati, Ludovica et al. "CAF-induced physical constraints controlling T cell state and localization in solid tumours." *Nature reviews. Cancer* vol. 24,10 (2024): 676-693. doi:10.1038/s41568-024-00740-4
17. Pei, Liping et al. "Roles of cancer-associated fibroblasts (CAFs) in anti- PD-1/PD-L1 immunotherapy for solid cancers." *Molecular cancer* vol. 22,1 29. 10 Feb. 2023, doi:10.1186/s12943-023-01731-z
18. Mantovani, Alberto et al. "Macrophages as tools and targets in cancer therapy." *Nature reviews. Drug discovery* vol. 21,11 (2022): 799-820. doi:10.1038/s41573-022-00520-5
19. Lu, Yiming et al. "A single-cell atlas of the multicellular ecosystem of primary and metastatic hepatocellular carcinoma." *Nature communications* vol. 13,1 4594. 6 Aug. 2022, doi:10.1038/s41467-022-32283-3
20. Cai, Ning et al. "Targeting MMP9 in CTNNB1 mutant hepatocellular carcinoma restores CD8+ T cell-mediated antitumour immunity and improves anti-PD-1 efficacy." *Gut* vol. 73,6 985-999. 10 May. 2024, doi:10.1136/gutjnl-2023-331342
21. Ngiow, Shin Foong et al. "LAG-3 sustains TOX expression and regulates the CD94/NKG2-Qa-1b axis to govern exhausted CD8 T cell NK receptor expression and cytotoxicity." *Cell* vol. 187,16 (2024): 4336-4354.e19. doi:10.1016/j.cell.2024.07.018
22. Miller, Brian C et al. "Subsets of exhausted CD8+ T cells differentially mediate tumor control and respond to checkpoint blockade." *Nature immunology* vol. 20,3 (2019): 326-336. doi:10.1038/s41590-019-0312-6
23. Paz-Ares, Luis et al. "First-line nivolumab plus ipilimumab combined with two cycles of chemotherapy in patients with non-small-cell lung cancer (CheckMate 9LA): an international, randomised, open-label, phase 3 trial." *The Lancet. Oncology* vol. 22,2 (2021): 198-211. doi:10.1016/S1470-2045(20)30641-0
24. Schürch, Christian M et al. "Coordinated Cellular Neighborhoods Orchestrate Antitumoral Immunity at the Colorectal Cancer Invasive Front." *Cell* vol. 182,5 (2020): 1341-1359.e19. doi:10.1016/j.cell.2020.07.005

Reviewer #4 (Remarks to the Author):

The core breakthrough of the article is the first realization of predicting the spatial expression of 40 proteins from a single H&E section, which solves the key bottlenecks of spatialomics for clinical translation - cost and accessibility. The proposed MICA multimodal fusion framework is also a novel solution. However, the differences with spatial transcriptome prediction efforts could be discussed in more depth, with existing comparisons only briefly mentioning performance differences. No analysis shows how HEX-MICA changes treatment decisions (e.g., % of patients reclassified vs. standard care) or improves patient outcomes. Without evidence of clinical utility beyond statistical metrics, the translational impact remains speculative.

Response: Thank you for recognizing the strengths and novelty of our work. We sincerely appreciate your comment on how the HEX-MICA framework could change treatment decisions, which we agree is an important point to demonstrate clinical impact. We have addressed this in detail in our response to your comment #4 below.

Here are the issues

1. The training set (755K image blocks) and validation set (372 samples) were adequately sized, and the multicenter cohort design (6 cohorts/2,298 patients) significantly enhanced conclusion reliability. However, specific margins of error for H&E-CODEX alignment and quality control processes (e.g., PALOM tool parameters mentioned in Page 15) need to be described.

Response: Thank you for raising this point. To quantify the spatial accuracy of H&E-CODEX alignment, we performed a cross-modality validation using independent cell segmentations from each image modality. Specifically, we segmented all cells on the original CODEX and H&E WSIs from Stanford-WSI using modality-specific pipelines, then applied the PALOM-derived transformation to map CODEX cell centroids into the H&E coordinate frame. For each cell in the H&E image, we computed the Euclidean distance to the nearest mapped CODEX cell. We also measured the diameter of each cell nucleus as a biological reference scale. As shown in Extended Data Fig. 39, 99.2% of cell-cell distances between H&E and CODEX were smaller than the median cell nucleus diameter, indicating that accurate cross-modality alignment is achieved with a subcellular resolution. We have included both the alignment accuracy results and key PALOM parameters—such as level= 0 alignment, 4,000-keypoint affine initialization, and block-wise shift refinement—in the revised Methods to clarify the registration procedure and quality controls.

Extended Data Fig. 39 | Validation of H&E-CODEX registration accuracy. Left: Violin plots showing the distribution of minimum distances between H&E and mapped CODEX cell centroids compared to CODEX cell diameters. Right: Representative H&E image with mapped CODEX nuclei centroids overlaid in yellow.

Revised Manuscript:

Methods - CODEX and H&E image registration and dataset construction (lines 673-675, 687-695)

2. Existing results should be compared with spatial transcriptome prediction studies from other studies to highlight protein stability and clinical applicability.

Response: Thank you for this valuable suggestion. To better contextualize HEX in relation to spatial transcriptomics (ST)-based models, we performed additional experiments comparing both biomarker prediction accuracy and downstream clinical applications.

First, we compared HEX against state-of-the-art ST prediction methods, including Omic-CLIP [1] and BLEEP [2], which are trained and evaluated on a public lung cancer dataset with 20 paired Visium ST and H&E slides [3]. The ST models were trained to predict the 2,000 highly variable genes, and we benchmarked prediction accuracy using Pearson and Spearman correlations. HEX achieved substantially higher performance for the full set of markers as well as the 18 genes overlapping with the protein targets (Extended Data Fig. 35). These findings demonstrate that protein prediction from H&E yields a much higher accuracy than transcript-level prediction, likely due to the improved stability of protein targets in clinical samples and their more direct relations to morphological phenotypes and biological functions.

Second, we assessed clinical utility of both approaches by applying the same multimodal integration framework (MICA) to two input combinations: H&E plus virtual ST

versus H&E plus virtual CODEX. In the Stanford-IO cohort, MICA-CODEX significantly outperformed MICA-ST (2,000 genes) for predicting objective response (AUC = 0.82 vs. 0.75, $P < 0.001$) as well as progression-free survival (C-index = 0.72 vs. 0.65, $P < 0.001$; Extended Data Fig. 36).

Taken together, these results demonstrate that HEX not only yields more accurate spatial proteomic maps than transcriptomics-based approaches, but also provides greater clinical utility in predicting treatment outcomes.

Extended Data Fig. 35 | Comparison of prediction accuracy between HEX and spatial transcriptomics (ST)-based models. Violin plots showing Pearson (left) and Spearman (right) correlations of predicted molecular profiles by HEX, BLEEP, and OmiCLIP.

Extended Data Fig. 36 | Comparison of clinical utility between virtual spatial transcriptomics and virtual proteomics. Performance of MICA models trained on H&E combined with virtual ST or virtual CODEX in the Stanford-IO cohort. Left: immunotherapy response prediction (AUC). Right: progression-free survival prediction (C-index). MICA-CODEX significantly outperformed MICA-ST in both tasks ($P < 0.001$). Error bars denote 95% CIs.

Revised Manuscript:

Results - Comparison with spatial transcriptomics-based prediction (lines 456-474)

Methods - Comparison with spatial transcriptomics prediction models (lines 943-954)

3. Why did the authors choose antibodies to the above 40 proteins, and are they specific enough for tumor marker.

Response: Thank you for this question. Regarding the marker selection, the 40 protein markers were chosen based on extensive prior literature and their well-established clinical and biological relevance in the lung tumor microenvironment (TME). The panel was designed to comprehensively capture key cellular populations, functional states, and microenvironmental remodeling features in the TME, and includes the following categories:

1. Lineage and structural markers (e.g., Pan-Cytokeratin, EpCAM, E-Cadherin, CD45, CD3E)
2. Functional and activation markers (e.g., Ki67, HIF1A)
3. Immune lineage subtype markers (e.g., CD4, CD8, FOXP3, CD163, CD68, CD66b)
4. Stromal and vascular markers (e.g., FAP, α SMA, Collagen IV, CD31)
5. Immune checkpoint and exhaustion markers (e.g., PD-1, PD-L1, ICOS, LAG-3, VISTA, CD39)
6. Tumor microenvironment remodeling markers (e.g., MMP9)
7. Antigen presentation markers (e.g., HLA-A, HLA-E, HLA-DR)

This antibody panel enables comprehensive spatial profiling of the epithelial, immune, and stromal components in the TME, supporting both phenotypic annotation and downstream functional interpretation within the HEX framework. Notably, the inclusion of immune lineage markers, as well as immune checkpoint and exhaustion markers in the panel, strengthens the HEX's capacity to enhance the prediction immunotherapy response.

Revised Manuscript:

Methods - CODEX with high-plex PhenoCycler imaging - Marker panel design (lines 633-638)

4. In the section on HEX improves immunotherapy response prediction, the authors only state that the system can predict a patient's immune sensitivity, but do not further explore how patient survival can be improved by this system.

Response: Thank you for raising this important point. To better contextualize how the HEX model can influence clinical decision-making, we conducted additional analyses beyond conventional performance metrics.

First, we assessed its ability to improve patient stratification for immunotherapy response. Compared to standard biomarker PD-L1, the HEX model correctly reclassified 4 out of 35 (11%) patients as true responders; at the same time, HEX also correctly reclassified 6 out of 55 (11%) patients as true non-responders, resulting in a positive Net Reclassification Index of 22%.

Second, we compared the HEX model to standard biomarkers including PD-L1 and TMB in terms of sensitivity at a fixed specificity of 90% for predicting objective response. Our model achieved a sensitivity of 61%, substantially outperforming PD-L1 (19%) and TMB (10%) (Extended Data Fig. 30). This analysis underscores the potential of HEX model to better distinguish true responders from non-responders, thereby improving the effective response rates while sparing patients of unnecessary treatment.

We then assessed how the new model would improve treatment decisions in the context of current clinical practice. ICI monotherapy is a standard of care in NSCLC patients with tumor PD-L1 $\geq 50\%$ given the likelihood of strong response. However, our HEX model identified 19 out of 47 (40%) patients with PD-L1-high tumors who did not experience durable response to ICI, with a median PFS interval of 4 months vs. 20 months in the predicted responder group (Fig. 6c). These patients may benefit from concurrent chemotherapy in addition to ICI. For patients with intermediate levels of PD-L1 ($>1\%$ and $<50\%$), the HEX model significantly stratified patients for PFS (HR = 3.57, P = 0.0009) and thus could also help optimize the treatment decisions on ICI monotherapy vs. combination ICI and chemotherapy.

On the other hand, for patients with tumor PD-L1 = 0%, there is a moderate benefit of anti-CTLA/anti-PD1 ICIs and chemotherapy over standard anti-PD1 ICIs and chemotherapy [4]. However, the triplet combination therapy has increased toxicity, and the optimal regimen is unclear at the patient level. The HEX model identified 23 out of 38 (61%) patients with PD-L1-negative tumors who did not benefit from anti-PD1 ICIs, with a median PFS interval of 4 months vs. 13 months in the predicted responder group (Fig. 6c). These patients may benefit from novel combination treatment strategies.

Finally, patients with EGFR-mutant tumors are generally considered refractory to ICIs and therefore do not receive ICIs as their standard treatment. In our analysis of the immunotherapy cohort, the HEX model identified 5 out of 17 (29%) of patients whose tumors harbor EGFR mutation but derived durable clinical benefit from ICIs, with a median PFS interval of 8 months vs. 4 months in the predicted non-responder group (Extended Data Fig. 31).

Taken together, these results suggest that virtual spatial proteomics and multimodal modeling with HEX–MICA can provide meaningful improvements in patient stratification, with the potential to enhance treatment selection and improve patient outcomes.

Extended Data Fig. 30 | Sensitivity at fixed specificity for predicting immunotherapy response. Comparison of MICA, PD-L1, and TMB for predicting objective response at 90% specificity.

Extended Data Fig. 31 | Kaplan–Meier analysis of MICA risk groups in Stanford-IO subgroups defined by EGFR. P values were calculated using two-sided log-rank tests.

Revised Manuscript:

Results - HEX improves immunotherapy response prediction in advanced lung cancer (lines 394-408)

Discussion (lines 541-557)

5. Test HEX on non-NSCLC samples (e.g., CRC/breast cancer) in supplementary analyses to assess pan-cancer utility.

Response: Thank you for raising this important question. To evaluate the broader applicability of HEX beyond lung cancer, we have conducted a series of experiments

across multiple datasets, cancer types, and different biomarker panels to assess the generalizability of HEX.

1. **External validation across diverse tissues**

We directly applied the NSCLC-trained HEX model—without any retraining or fine-tuning—to a pan-cancer dataset with paired 57-plex CODEX and H&E images from the University of Bern. This publicly available dataset includes 206 TMA cores spanning 34 tissue types, including malignant tumors (e.g., breast, colorectal, stomach, liver, pancreas, kidney, bladder, etc.), benign neoplasms, and normal tissues, prepared using different staining protocols and scanned on a different imaging platform [24]. Despite these technical variations, HEX achieved strong protein prediction performance across 24 overlapping biomarkers (mean Pearson $r = 0.658$, Extended Data Fig. 5), only slightly lower than on the original Stanford dataset ($r = 0.718$). Further, HEX substantially outperformed the second-best model, CGAN, consistently across all key metrics: Pearson r (0.658 vs. 0.210), Spearman r (0.563 vs. 0.140), SSIM (0.638 vs. 0.521), and MSE (0.132 vs. 0.835), confirming its generalizability across tissue types, staining protocols, and imaging platforms.

2. **Extension to new protein markers in a different cancer**

To further test adaptability of HEX to new marker panels in other tissues, we performed additional experiments by retraining or finetuning the HEX model to predict all 57 protein markers—including 33 not present in the original NSCLC panel. For this purpose, we used 140 colorectal cancer (CRC) cores from the Bern dataset. When trained de novo, HEX achieved a Pearson correlation of $r = 0.566$ (Extended Data Fig. 6). Fine-tuning the NSCLC-trained model by reinitializing the output head and updating all parameters improved performance to Pearson correlation $r = 0.659$ (Extended Data Fig. 7). These improvements were consistent across both legacy and CRC-specific markers (Extended Data Fig. 8), demonstrating that HEX can be efficiently adapted to other tissue types and expanded marker panels with limited additional training data.

3. **Pan-cancer prognosis prediction using virtual spatial proteomics**

To evaluate clinical utility across cancer types, we extended prognosis prediction to 12 additional cancer types involving 5,019 patients in TCGA. Using virtual proteomic maps generated by HEX, we trained the multimodal MICA model and compared its prognostic performance to two unimodal baselines (H&E-only and HEX-derived virtual CODEX). MICA consistently outperformed both baselines (mean c-index = 0.732, Extended Data Fig. 25), with significant stratification across all cancer types (log-rank $P \leq 0.0001$). These results confirm that HEX-derived features contribute to clinically meaningful predictions across histologically diverse malignancies.

Taken together, these additional analyses demonstrate that HEX is broadly generalizable across tissue types, cancer types, and biomarker panels, and can be efficiently adapted to new clinical settings with limited training data.

Extended Data Fig. 5 | External validation of HEX on the Bern pan-cancer dataset. a, Pearson correlation between predicted and measured protein expression across 24 overlapping markers. **b,** Comparison of HEX performance between two imaging platforms: Leica AT2 (Stanford-TMA and TA-TMA) and Keyence BZ-X710 (Bern). **c,** Comparison of HEX with baseline models on the Bern dataset. Error bars indicate 95% CIs from 1,000 bootstrap replicates.

Extended Data Fig. 6 | HEX retrained on colorectal cancer. Pearson correlation for 57 markers after retraining HEX on CRC. Orange bars denote markers overlapping with the NSCLC panel; blue bars are CRC-specific. Error bars show 95% CIs.

Extended Data Fig. 7 | HEX fine-tuned on colorectal cancer. Pearson correlation for 57 markers after fine-tuning the NSCLC-trained HEX model on CRC. Orange bars indicate markers overlapping with the NSCLC panel; blue bars are CRC-specific. Error bars show 95% CIs.

Extended Data Fig. 8 | Fine-tuning improves prediction across legacy and novel markers. Performance comparison on legacy (24) and novel (33) markers after retraining or fine-tuning. Error bars show 95% CIs.

Extended Data Fig. 25 | MICA improves pan-cancer prognosis prediction. a, C-index comparison of H&E-only (MUSK), virtual CODEX-only, and MICA models. MICA outperforms both baselines across all cancer types. Error bars show standard deviation. **b**, Kaplan–Meier curves for 12 TCGA cancer types using MICA-predicted risk groups.

Revised Manuscript:

Results - HEX enables accurate prediction of protein expression from H&E images - Independent validation performance (lines 169-197)

Results - HEX improves pan-cancer prognosis prediction (lines 289-305)

Methods - Training and evaluation of MICA (lines 892-898, 907-910)

6. "Virtual proteomics" are never validated with orthogonal methods (e.g., IHC, CyTOF). The study relies entirely on predicted protein patterns to derive biological insights (e.g., Fig 5c-d, Fig 6d-e). Correlations may reflect algorithmic artifacts rather than true biology.

Response: Thank you for raising these points. To provide orthogonal validation of HEX predictions, we evaluated the model on three lung samples from the publicly available ANHIR challenge dataset (<https://anhir.grand-challenge.org/Data/>), which includes whole-slide IHC images for CD31 and Ki67 stains obtained from adjacent 3 μm FFPE sections [6]. We directly applied HEX to this dataset without any retraining or fine-tuning. Although the IHC and H&E slides were not from the same tissue section, HEX was still able to generate virtual protein maps that quantitatively match well with the IHC data. Specifically, HEX predicted IHC measurements from H&E with strong spatial concordance—Pearson $r = 0.479$ (CD31) and 0.606 (Ki67), Spearman $r = 0.468$ and 0.590 —and substantially outperformed the CGAN model (Pearson: 0.542 vs 0.086 ; Spearman: 0.529 vs 0.057), as shown in Extended Data Fig. 9. These results provide orthogonal, modality-independent support for the utility of HEX-derived virtual spatial proteomics, even under the challenge of adjacent-section alignment and cross-modality variations.

Extended Data Fig. 9 | Orthogonal validation of HEX-derived protein expression using IHC. Comparison of HEX with two generative AI models using Pearson (left) and Spearman (right) correlation coefficients for CD31 and Ki67 across three lung samples from the ANHIR dataset. Error bars show 95% CIs.

Regarding your question on biological validation, we indeed validated the candidate spatial proteomic markers for prognostic prediction using true CODEX data. Please refer to **Figure 5e-f**. To assess the biological interpretation and relevance of HEX-derived predictions of immune checkpoint therapy response, we focused on cellular subsets which are defined by a lineage-defining marker with a functional or state-associated marker. While not experimentally validated in this study, below we briefly

outline the mechanistic evidence and biological relevance of HEX-nominated candidate cell types which are supported by functional experiments conducted in prior studies.

1. Granzyme B⁺ CD8⁺ T cells: These represent activated cytotoxic T lymphocytes (CTLs), Granzyme B is a key effector molecule delivered via perforin-mediated pores to induce apoptosis in antigen-expressing tumor cells. Tumor cell killing, cytolytic granule release, and its correlation with immune checkpoint therapy response have been well documented through functional experiments [7, 8, 9].

2. TCF-1⁺ CD4⁺ T cells: This population represents naïve or stem-like helper T cells that exhibit memory-like or progenitor characteristics and are capable of sustaining long-term immunity under chronic antigenic stimulation. These cells have been shown to preserve proliferative capacity and contribute to the maintenance of functional CD8⁺ T cell responses in both autoimmunity and cancer. Their relevance to immunotherapy efficacy has been demonstrated in preclinical tumor models and clinical cohorts [10]. Furthermore, the *Tcf7* locus, which encodes TCF-1, has been shown to undergo de novo epigenetic reprogramming in circulating autoimmune CD4⁺ T cells, supporting its role as a developmentally pre-programmed regulator of long-lived CD4⁺ T cell states [11]. These findings highlight the functional importance of TCF-1⁺ CD4⁺ T cells in both immune memory and therapeutic response settings.

3. MMP9⁺ CD66b⁺ neutrophils: This cellular subset corresponds to tumor-associated neutrophils (TANs) characterized by matrix remodeling and pro-tumorigenic properties. Functionally, MMP9 contributes to tumor progression by promoting angiogenesis, facilitating immune evasion, and supporting metastatic dissemination [12, 13, 14, 15, 16]. Recent studies have further identified MMP9⁺ CXCR2⁺ TANs as a distinct immunosuppressive state enriched in the tumor microenvironment of both human tumors and mouse models [17].

4. PD-1⁺ CD8⁺ T cells: These include exhausted or progenitor-like CD8⁺ T cells. Their ability to respond to checkpoint blockade is well documented, and their presence in tumors predicts therapeutic response depending on their differentiation state and TCF1 co-expression [18, 19].

5. FAP⁺ Collagen IV⁺ CAFs: This population represents matrix-producing cancer-associated fibroblasts (mCAFs) that play a central role in stromal remodeling and immune exclusion. These cells contribute to therapy resistance by constructing both physical extracellular matrix (ECM) barriers and establishing immunosuppressive chemical gradients that impede T cell infiltration [20, 21, 22]. Functional studies have demonstrated that FAP⁺ CAFs deposit dense collagen networks and modulate the

tumor microenvironment to favor immune evasion, thereby limiting the efficacy of immunotherapy [23].

6. MMP9⁺ CD163⁺ macrophages: This subset of tumor-associated macrophages (TAMs) exhibits potent matrix-remodeling and immunosuppressive activity. Through MMP9 secretion, a key member of the matrix metalloproteinase family, these cells promote extracellular matrix degradation and contribute to the formation of a suppressive tumor microenvironment that impedes effector T cell infiltration [24, 25]. Recent studies in CTNNB1-mutant hepatocellular carcinoma further demonstrate that pharmacological inhibition of MMP9 reactivates CD8⁺ T cell-mediated antitumor immunity and enhances the efficacy of anti-PD-1 therapy [26]. These findings underscore the mechanistic importance of MMP9⁺ TAMs in driving immune evasion and immunotherapy resistance.

Beyond the single-cell phenotypes described above, we computed a spatial proximity (Jaccard index) to quantify the pairwise cell-cell co-localization patterns and evaluate its association with immunotherapy response. For the cell-cell correlation matrix shown in Fig. 6e, several lines of evidence, including functional studies, support the biological plausibility of our findings. Specifically, the spatial proximity between PD1⁺ CD8⁺ T cells and Granzyme B⁺ CD8⁺ T cells was significantly elevated in the responders vs non-responders, suggesting that co-localization of exhausted or progenitor-like CD8⁺ T cells with cytotoxic effector CD8⁺ T cells may reflect an ongoing phenotypic transition within the tumor microenvironment. This spatial organization could enable effective reinvigoration of T cell cytotoxicity upon immune checkpoint blockade. Prior studies have shown that PD1 blockade promotes the differentiation of PD1⁺ progenitor CD8⁺ T cells into functional cytotoxic lymphocytes, thereby improving therapeutic efficacy [27, 28].

Overall, these findings support that the cell phenotypes implicated by our HEX framework are independently validated through in vivo and in vitro functional experiments. Moreover, the spatial cell-cell interactions uncovered by HEX are experimentally supported and mechanistically linked to immunotherapy response and clinical outcomes. We have discussed this in the revised Discussion section.

Revised Manuscript:

Results - HEX enables accurate prediction of protein expression from H&E images - Independent validation performance (lines 199-206)

Methods - WSI processing (lines 751-754)

Discussion (lines 564-577)

Reference (Reviewer #4):

1. Chen, Weiqing, et al. "A visual-omics foundation model to bridge histopathology with spatial transcriptomics." *Nature Methods* (2025): 1-15.
2. Xie, Ronald, et al. "Spatially resolved gene expression prediction from histology images via bi-modal contrastive learning." *Advances in Neural Information Processing Systems* 36 (2023): 70626-70637.
3. Madissoon, Elo, et al. "A spatially resolved atlas of the human lung characterizes a gland-associated immune niche." *Nature genetics* 55.1 (2023): 66-77.
4. Paz-Ares, Luis et al. "First-line nivolumab plus ipilimumab combined with two cycles of chemotherapy in patients with non-small-cell lung cancer (CheckMate 9LA): an international, randomised, open-label, phase 3 trial." *The Lancet. Oncology* vol. 22,2 (2021): 198-211. doi:10.1016/S1470-2045(20)30641-0
5. Schürch, Christian M et al. "Coordinated Cellular Neighborhoods Orchestrate Antitumoral Immunity at the Colorectal Cancer Invasive Front." *Cell* vol. 182,5 (2020): 1341-1359.e19. doi:10.1016/j.cell.2020.07.005
6. Borovec, Jiri et al. "ANHIR: Automatic Non-Rigid Histological Image Registration Challenge." *IEEE transactions on medical imaging* vol. 39,10 (2020): 3042-3052. doi:10.1109/TMI.2020.2986331
7. St Paul, Michael, and Pamela S Ohashi. "The Roles of CD8+ T Cell Subsets in Antitumor Immunity." *Trends in cell biology* vol. 30,9 (2020): 695-704. doi:10.1016/j.tcb.2020.06.003
8. Halle, Stephan et al. "In Vivo Killing Capacity of Cytotoxic T Cells Is Limited and Involves Dynamic Interactions and T Cell Cooperativity." *Immunity* vol. 44,2 (2016): 233-45. doi:10.1016/j.immuni.2016.01.010
9. Weigelin, Bettina et al. "Cytotoxic T cells are able to efficiently eliminate cancer cells by additive cytotoxicity." *Nature communications* vol. 12,1 5217. 1 Sep. 2021, doi:10.1038/s41467-021-25282-3
10. Zou, Dawei et al. "CD4+ T cell immunity is dependent on an intrinsic stem-like program." *Nature immunology* vol. 25,1 (2024): 66-76. doi:10.1038/s41590-023-01682-z
11. Aljobaily, Nouf et al. "Autoimmune CD4+ T cells fine-tune TCF1 expression to maintain function and survive persistent antigen exposure during diabetes." *Immunity* vol. 57,11 (2024): 2583-2596.e6. doi:10.1016/j.immuni.2024.09.016
12. Ardi, Veronica C et al. "Human neutrophils uniquely release TIMP-free MMP-9 to provide a potent catalytic stimulator of angiogenesis." *Proceedings of the National Academy of Sciences of the United States of America* vol. 104,51 (2007): 20262-7. doi:10.1073/pnas.0706438104

13. Vannitamby, Amanda et al. "Tumour-associated neutrophils and loss of epithelial PTEN can promote corticosteroid-insensitive MMP-9 expression in the chronically inflamed lung microenvironment." *Thorax* vol. 72,12 (2017): 1140-1143. doi:10.1136/thoraxjnl-2016-209389
14. Coussens, L M et al. "MMP-9 supplied by bone marrow-derived cells contributes to skin carcinogenesis." *Cell* vol. 103,3 (2000): 481-90. doi:10.1016/s0092-8674(00)00139-2
15. Zilionis, Rapolas et al. "Single-Cell Transcriptomics of Human and Mouse Lung Cancers Reveals Conserved Myeloid Populations across Individuals and Species." *Immunity* vol. 50,5 (2019): 1317-1334.e10. doi:10.1016/j.immuni.2019.03.009
16. Xue, Ruidong et al. "Liver tumour immune microenvironment subtypes and neutrophil heterogeneity." *Nature* vol. 612,7938 (2022): 141-147. doi:10.1038/s41586-022-05400-x
17. Wu, Yingcheng et al. "Neutrophil profiling illuminates anti-tumor antigen-presenting potency." *Cell* vol. 187,6 (2024): 1422-1439.e24. doi:10.1016/j.cell.2024.02.005
18. Humblin, Etienne et al. "The costimulatory molecule ICOS limits memory-like properties and function of exhausted PD-1+CD8+ T cells." *Immunity*, S1074-7613(25)00248-1. 1 Jul. 2025, doi:10.1016/j.immuni.2025.06.001
19. Damo, Martina et al. "PD-1 maintains CD8 T cell tolerance towards cutaneous neoantigens." *Nature* vol. 619,7968 (2023): 151-159. doi:10.1038/s41586-023-06217-y
20. Cords, Lena et al. "Cancer-associated fibroblast phenotypes are associated with patient outcome in non-small cell lung cancer." *Cancer cell* vol. 42,3 (2024): 396-412.e5. doi:10.1016/j.ccell.2023.12.021
21. Forsthuber, Agnes et al. "Cancer-associated fibroblast subtypes modulate the tumor-immune microenvironment and are associated with skin cancer malignancy." *Nature communications* vol. 15,1 9678. 8 Nov. 2024, doi:10.1038/s41467-024-53908-9
22. Arpinati, Ludovica et al. "CAF-induced physical constraints controlling T cell state and localization in solid tumours." *Nature reviews. Cancer* vol. 24,10 (2024): 676-693. doi:10.1038/s41568-024-00740-4
23. Pei, Liping et al. "Roles of cancer-associated fibroblasts (CAFs) in anti- PD-1/PD-L1 immunotherapy for solid cancers." *Molecular cancer* vol. 22,1 29. 10 Feb. 2023, doi:10.1186/s12943-023-01731-z
24. Mantovani, Alberto et al. "Macrophages as tools and targets in cancer therapy." *Nature reviews. Drug discovery* vol. 21,11 (2022): 799-820. doi:10.1038/s41573-022-00520-5
25. Lu, Yiming et al. "A single-cell atlas of the multicellular ecosystem of primary and metastatic hepatocellular carcinoma." *Nature communications* vol. 13,1 4594. 6 Aug. 2022, doi:10.1038/s41467-022-32283-3
26. Cai, Ning et al. "Targeting MMP9 in CTNNB1 mutant hepatocellular carcinoma restores CD8+ T cell-mediated antitumour immunity and improves anti-PD-1 efficacy." *Gut* vol. 73,6 985-999. 10 May. 2024, doi:10.1136/gutjnl-2023-331342

27. Ngiow, Shin Foong et al. "LAG-3 sustains TOX expression and regulates the CD94/NKG2-Qa-1b axis to govern exhausted CD8 T cell NK receptor expression and cytotoxicity." *Cell* vol. 187,16 (2024): 4336-4354.e19. doi:10.1016/j.cell.2024.07.018
28. Miller, Brian C et al. "Subsets of exhausted CD8+ T cells differentially mediate tumor control and respond to checkpoint blockade." *Nature immunology* vol. 20,3 (2019): 326-336. doi:10.1038/s41590-019-0312-6